# An integrative epigenome-based strategy for unbiased functional profiling of clinical kinase inhibitors

Francesco Gualdrini [ID] [1,✉], Stefano Rizzieri[1,2], Sara Polletti [ID] [1,2], Francesco Pileri[1,2], Yinxiu Zhan [ID] [1], Alessandro Cuomo [ID] [1] & Gioacchino Natoli [ID] [1,✉]

## Abstract

**More than 500 kinases are implicated in the control of most cellular process in mammals, and deregulation of their activity is linked to cancer and inflammatory disorders. 80 clinical kinase inhibitors (CKIs) have been approved for clinical use and hundreds are in various stages of development. However, CKIs inhibit other kinases in addition to the intended target(s), causing both enhanced clinical effects and undesired side effects that are only partially predictable based on in vitro selectivity profiling. Here, we report an integrative approach grounded on the use of chromatin modifications as unbiased, information-rich readouts of the functional effects of CKIs on macrophage activation. This approach exceeded the performance of transcriptome-based approaches and allowed us to identify similarities and differences among CKIs with identical intended targets, to recognize novel CKI specificities and to pinpoint CKIs that may be repurposed to control inflammation, thus supporting the utility of this strategy to improve selection and use of CKIs in clinical settings.**

**Keywords** Clinical Kinase Inhibitors; Inflammation; Machine Learning; Epigenome; Drug Repurposing
**Subject Categories** Chromatin, Transcription & Genomics; Methods & Resources

## Introduction

As phospho-transfer reactions occur in virtually every signal transduction process, kinases stand out as central regulators of normal and pathological cell functions. The human genome contains 518 genes encoding for a kinase domain (Manning et al, 2002) and 510 of them have a highly conserved mouse ortholog (Caenepeel et al, 2004). These kinases account for the many thousands of phospho-sites identified in human and mouse cells and currently annotated in databases (Hornbeck et al, 2015),

implying that the signaling scope of individual kinases as well as the complexity of cross-talks among signaling pathways inside cells remain vastly under-characterized.

Given that protein kinases regulate nearly all aspects of cell life, inhibition of kinases has been extensively pursued by pharmaceutical companies over the last 30 years. 80 kinase inhibitors (KIs) are approved for clinical use (Roskoski, 2024) and hundreds more are being tested (Carles et al, 2018) (http://www.icoa.fr/pkidb/). While most clinical KIs (CKIs) are employed in the treatment of various cancers, some of them are currently used in the therapy of non-malignant diseases such as autoimmune and inflammatory disorders (Ferguson and Gray, 2018), and broad opportunities exist for the use of CKIs for the treatment of other diseases (Cohen and Alessi, 2013; Zarrin et al, 2021).

A few CKIs, such as the MEK1/2 inhibitor Trametinib (Abe et al, 2011), interact with surfaces outside of the ATP-binding site and act as allosteric inhibitors claimed to possess relatively high specificity for their intended target(s). However, most CKIs act by competing with the conserved ATP-binding site in the kinase domain and thus show cross-activity on multiple kinases. Overall, clinically approved CKIs do not appear to have higher specificity compared to compounds in earlier phases of development (Klaeger et al, 2017), implying that selectivity does not increase the chances of clinical approval. In fact, simultaneous inhibition of multiple kinases and downstream effectors may enhance therapeutic effects; however, non-selective inhibition of multiple kinases is also expected to increase risk and extent of side effects.

Aside from its implications in the delicate balance between therapeutic efficacy *vs*. side effects due to off-targeting, the polypharmacology of CKIs would benefit from the development of profiling approaches that accurately describe their composite effects in living cells. However, CKI selectivity profiling is typically carried out using in vitro assays. Traditional approaches to profile CKI selectivity using kinase assays with recombinant kinases have been largely superseded by assays measuring the ability of CKIs to competitively displace kinases pulled down from cell lysates using ATP analogs-coated beads (kinobeads) (Bantscheff and Drewes, 2012; Bantscheff et al, 2007). Kinobeads-based assays have two main advantages: first, they allow profiling binding selectivity of CKIs towards endogenously expressed kinases rather than

[1]Department of Experimental Oncology, IEO, European Institute of Oncology IRCCS, Milano 20139, Italy. [2]These authors contributed equally: Stefano Rizzieri, Sara Polletti, Francesco Pileri. ✉E-mail: francesco.gualdrini@ieo.it; gioacchino.natoli@ieo.it

recombinant kinase domains; and second, they generate a measure of binding selectivity that integrates the two essential determinants of CKI activity, namely the intrinsic affinity of CKIs towards the target kinase(s) and the affinity of target kinases towards ATP. However, only a fraction of the kinome can be pulled down using kinobeads (Eberl et al, 2019; Klaeger et al, 2017) and since it is an in vitro assay based exclusively on the measurement of binding-unbinding events, its ability to predict cellular effects of CKIs is unclear. Alternative chemoproteomics approaches based on the use of biotin-labeled CKIs on cell lysates followed by mass spectrometry have been reported (Patricelli et al, 2011), with advantages and limitations similar to those of kinobeads. Other techniques that measure selectivity profiles in a more physiological setting include assays based on the effects of small molecule binding on the thermal stability of their targets in cells or cell lysates (Savitski et al, 2014). These assays, however, have a relatively limited throughput and they may have detection limits in the case of proteins expressed at very low levels. Moreover, the inherent complexities of drug-protein interactions in cells can result in a broad panel of direct and indirect effects on thermal stability of a given drug target, ranging from increased stability to no changes and even reduced thermal stability (Seashore-Ludlow et al, 2020), thus complicating data interpretation. Finally, recent technical advancements enabled the generation of global phosphoproteomics maps to delve deeper into the dynamic cellular responses to CKIs (Lee et al, 2024), and specific algorithms have been developed that aim to reconstruct topological relationships among protein-kinases and their signaling networks (Hijazi et al, 2020). By examining the effects of small molecule binding on the phosphoproteome and inferring enzyme-substrate relationships within a pathway, these methods provide the essential grounds to understand the impact of CKIs on intracellular signaling. On the other hand, however, they fall short in providing insights into the functional effects of CKIs within cells. Hence, they would ideally require integration with complementary assays providing functional readouts.

Based on these considerations, we set out to develop an in vivo experimental and analytical approach that would complement current selectivity profiling methods by providing high-content functional information on the actual effects of CKIs on cells using a specific and interpretable readout. We reasoned that although not all signaling events triggered by kinase activation result in transcriptional changes, a large number of them are indeed relayed to transcription factors (TFs) and additional nuclear regulators, eventually impacting TF-instructed deposition and removal of dynamic histone modifications such as histone acetylation. Histone acetylation changes occurring at hundreds or thousands of genomic regions in response to CKIs can then be analyzed through established statistical approaches to determine the TF DNA-binding motifs over-represented in the affected regions. This would in principle allow determining which combinations of TFs are impacted by each CKI, and thus, albeit indirectly, the upstream signaling pathway(s) regulated by each given inhibitor.

As a biological setting of biomedical relevance and therapeutic interest, we focused on the activation of the inflammatory gene expression programs in macrophages (Xue et al, 2014). Upon innate immune receptor triggering (Janeway and Medzhitov, 2002; Thaiss et al, 2016), signals transduced by multiple kinases (Arthur and Ley, 2013; Karin and Ben-Neriah, 2000; O'Shea et al, 2015; Sharma et al, 2003) activate transcription factors (TFs) such as NF-kB, IRF, STAT, and AP-1 family members that are combinatorially recruited with complex temporal kinetics to thousands of enhancers that control the expression of genes involved in inflammation and its resolution and whose acetylation rapidly and dynamically changes over time (Glass and Natoli, 2016). Specifically, we comparatively characterized the effects of a large panel of CKIs on two pathways triggering functionally divergent macrophage activation programs: the TLR4 (Toll-like Receptor 4) pathway, which is triggered by lipopolysaccharide (LPS) and leads to the generation of classical inflammatory macrophages (Fitzgerald and Kagan, 2020); and the Interleukin 4 (IL-4) signaling pathway (Junttila, 2018), which activates gene expression programs involved in immunity against parasites and in the resolution phase of acute inflammation (Fig. 1A, top left cartoon).

In addition to the activation of the MAP kinases and IkB kinases (IKK), which activate Rel/NF-kB family transcription factors, TLR4 triggering results in the phosphorylation of the transcription factor IRF3 by TBK1 (Sharma et al, 2003) (Fig. 1A). IRF3 phosphorylation is required for the rapid induction of the Interferon b1 gene (*Ifnb1*), IFNβ release and autocrine and paracrine stimulation of the type I IFN receptor (IFNAR) on the cell surface. IFNAR triggering peaks at 2 h after LPS stimulation and activates the Janus kinases (JAK) TYK2 and JAK1, leading to the phosphorylation of the STAT1 and STAT2, which induce hundreds of IFN-stimulated genes (ISGs) (Toshchakov et al, 2002). ISGs represent a prominent and biomedically relevant fraction of the secondary, protein synthesis-dependent wave of LPS-induced transcription. IL-4 stimulation of the IL-4 receptor (IL-4R) results in a comparatively simpler signal transduction, with the activation of JAK1 and JAK3 mainly causing the phosphorylation of STAT6, the pivotal regulator of IL-4-induced gene expression (Junttila, 2018).

To analyze the effects of CKIs on macrophage responses to LPS and IL-4, we devised an approach grounded on the use of histone acetylation and an integrated analytical pipeline. This strategy allowed us to identify similarities and differences among CKIs with identical or overlapping intended target(s), to recognize novel CKI specificities and to pinpoint CKIs that based on their effects may be repurposed to control macrophage activation.

## Results

### Histone acetylation as unbiased and specific readout of CKIs' effects

We selected CKIs that target: (i) components of the TLR4 or IL-4R signaling pathways and (ii) protein-kinases not assigned to either pathway but transcriptionally induced in response to LPS or IL-4 stimulation, as determined by a time-resolved nascent RNA-seq analysis (from 0 to 4 h post-stimulation) (Fig. 1A; Dataset EV1).

A critical issue we had to deal with is the lack of uniformity of available information on the effective concentration and selectivity of individual CKIs. The use of "averaged" IC50 measurements from multiple non-homogeneous sources would have introduced confounding effects and uncontrollable biases. Similarly, employing IC50 measurements from different sources and techniques for various kinase inhibitors would have been entirely arbitrary. Therefore, in order to use a consistent criterion for the selection of the CKIs to be employed in our study, we exploited a single

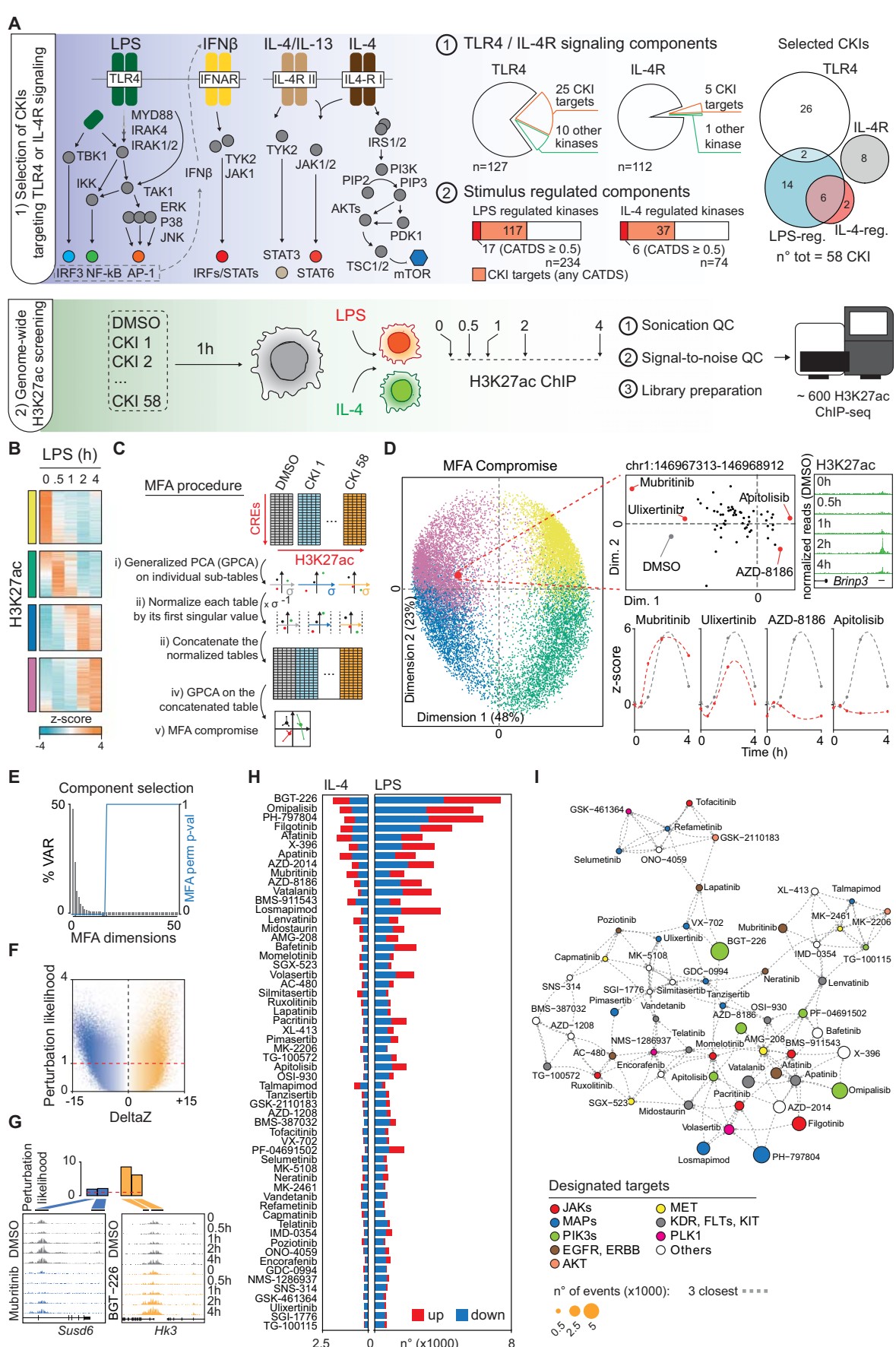

**Figure 1.  Determination of CKIs' effects in macrophages using epigenomic features.**

(A) Experimental outline. CKIs targeting signaling components of the TLR4 or the IL-4R pathway, or stimulus-regulated kinases were selected for a total of 58 CKIs. 25 TLR4 and 5 IL-4R components (top middle pie-charts) could be targeted with 28 and 8 CKIs, respectively (top right Venn-diagram), while 17 out of 117 LPS-regulated and 6 out of 37 IL-4-regulated genes could be targeted by 22 and 8 CKIs respectively (top right Venn-diagram). The ChIP-seq-based screening was carried out in primary mouse bone marrow-derived macrophages (BMDM) pre-treated with individual CKIs or vehicle (DMSO) for 1 h and then stimulated as indicated (bottom scheme). (B) Four main clusters of CREs displaying LPS-regulated H3K27ac are shown. The four groups of CREs are color-coded (the same color-code was used throughout all figures). (C) Schematic workflow of the MFA algorithm (adapted from (Abdi et al, 2013), which comprises: i. Generalized PCA (GPCA) of the 59 individual tables (58 CKIs and the DMSO); ii. Normalization of individual tables by the inverse of their first singular values; iii. Concatenation of individual tables; iv. GPCA of the concatenated table; v. Final MFA compromise derived from the final GPCA step. (D) Reduced dimensionality plot showing H3K27ac changes of individual CREs in macrophages stimulated with LPS. The right panel shows the first two dimensions of the MFA compromised analysis (de Tayrac et al, 2009; Escofier and Pagès, 2008) for the ~16,500 LPS-controlled CREs. Each CRE was colored according to the macro-clusters defined in panel (B). The position of each CRE in the MFA compromise plot corresponds to the average position of the same CRE considering all individual CKIs and the DMSO. A representative CRE located 5′ of the *Bripn3* gene and characterized by a peak of H3K27ac at 2 h post-LPS is shown as example. The position of this CRE in the 2D plot (inset) is the centroid of the 59 conditions tested (58 CKIs and the DMSO), each represented by a dot in the inset. The effects of four different CKIs on the H3K27ac profile of this specific CRE are also shown (bottom left). (E) MFA component selection via permutation testing (1000 iterations). Column-wise permutation was conducted to select components carrying variance more than by random. The gray bars (left y-axes) correspond to the observed unpermuted data tables; the blue line (right y-axes) correspond to the computed probability to observe equal or more variance by randomly column-wise permuted data tables. (F) Relationship between the computed perturbation likelihood score and the change in H3K27ac amplitude at each CRE. (G) Representative examples of perturbed CREs by the CKI Mubritinib and BGT-226. Top, perturbation likelihood bar chart. Bottom, individual H3K27ac tracks reporting the *Susd6* and *Hk3* loci (gray: DMSO, blue: Mubritinib; orange: BGT-226). (H) Number of identified changes per CKI with an associated perturbation likelihood score greater than 1, for each CKI in the two signaling context (right: LPS stimulations; left: IL-4 stimulations). In red are shown the events characterized by H3K27ac upregulation and in blue those characterized by H3K27ac downregulation. (I) Two-dimensional network graph (KNN computed on the basis of the perturbation likelihood score and the direction of change per CKI per CRE) showing the mutual relationships among CKIs. Each CKI is a vertex connected to the three closest CKIs. Each vertex was colored according to the designated target kinase family.

selectivity metric, the CATDS (concentration and target-dependent score), which is based on the measurement of the affinity of each CKI towards its target(s) in kinobeads assays (Klaeger et al, 2017). This score represents the reduction in binding of a specific protein kinase to kinobeads at a certain CKI concentration, divided by the total decrease in binding of all other target kinases. Therefore, CATDS scores are determined for each of the kinases targeted by a given CKI and they range from 0 (indicating the total lack of selectivity of the CKI) to 1 (indicating a CKI specifically targeting a single protein kinase). The use of kinobeads data for selection of CKIs is grounded on the notion that in vitro measured binding affinities correlate with the potency of CKIs in cells (Knight and Shokat, 2005). However, while unavoidable, this choice implies two main limitations. First, the repertoire of kinases assayed with kinobeads, albeit very broad, is only partial (253/518 kinases in the study by Klaeger et al) (Klaeger et al, 2017), with a large fraction of the kinome remaining undetectable due to sensitivity limits or other technical issues. Second, relying on a sole dataset for affinity measurements can introduce inherent measurement errors or biases into subsequent analyses. Importantly, because of the selection criteria used, some CKIs linked to a given kinase in clinical annotations were used to target a different kinase (heretofore indicated as the "*intended target*"). For instance, Bafetinib, an approved BCR-ABL and LYN tyrosine kinase inhibitor ($EC_{50}$ for ABL1 = 26.37 nM), was used here as a MAPKAPK3 inhibitor ($EC_{50}$ = 23.1 nM) and the PI3 kinase inhibitor Apitolisib was used to target JAK1 ($EC_{50}$ = 25.8 nM) (Appendix Fig. S1 and Dataset EV1).

Of the 127 TLR4 and 112 IL-4R signaling components annotated in the Reactome database (https://reactome.org), 35 and 6 are annotated kinases, respectively, and 30 of them (25 TLR4 and 5 IL-4R components) could be matched to 149 CKIs with a wide range of selectivity scores (Fig. 1A; Dataset EV1). To select a subset of CKIs that on the one hand possessed sufficient selectivity for TLR4 and IL-4R signaling components, and on the other some degree of target overlap (which is required for downstream

deconvolution, see below), we considered at most two CKIs per target, with a minimum selectivity score (CATDS) of 0.1. In total, we selected 36 CKIs targeting 24 TLR4 or IL-4R/IL-13R signaling components (19 TLR4 and 5 IL-4R components) (Fig. 1A, Appendix Fig. S1 and Dataset EV1). As regards stimulus-regulated kinases currently not linked to TLR4 or IL-4R signaling, 234 and 74 kinases were regulated at the transcriptional level by LPS or IL-4, respectively, and about half of them could be assigned to a total of 208 CKIs with a wide range of selectivity scores. Among these CKIs, we selected at most two compounds per target, with a CATDS ≥ 0.5, eventually resulting in a list of 24 inhibitors. Moreover, some CKIs can also engage ATP-binding enzymes other than kinases (Klaeger et al, 2017). Among them, NQO2 (N-Ribosyldihydronicotinamide:Quinone Reductase 2), TOP2B (DNA topoisomerase II beta), and ACOX3 (Acyl-CoA oxidase 3) were transcriptionally regulated by LPS or IL-4 and compounds targeting these enzymes were included in the list.

Overall, we selected a total of 58 CKIs whose specificity, properties, and clinical annotations are reported in Appendix Fig. S1 and Dataset EV1. Importantly, several of these CKIs target different members of the same kinase family (e.g., JAK family) or in some cases precisely the same kinase (e.g., Lenvatinib and Vandetanib were both used to target RIPK2), which in principle should result in extensive overlaps of effects. In addition, 238 pairs of CKIs shared more than one target, exhibiting different ranges of CATDS divergence. Hence, a substantial level of predicted target redundancy was deliberately present in the panel of selected inhibitors as it represented a valuable source of information (Appendix Fig. S1 and Dataset EV1). The intrinsic poly-pharmacology of each inhibitor has been strategically leveraged in the subsequent genomic analysis, as elucidated in detail below.

All CKIs were used at the half-maximal effective concentration ($EC_{50}$) towards the intended target, namely the concentration of the drug that displaces half of the intended target kinase from kinobeads (Klaeger et al, 2017). As the KIs used in our assay were meant for clinical use, they were optimized for permeability.

Nevertheless, given the potential discrepancies between in vitro measured $EC_{50}$ values and drug concentrations needed to bring about on-target effects in cells, we mined the available literature to collect the concentrations that for each CKI were previously experimentally tested to produce detectable effects in living cells (such as changes in phosphorylation events or more general functional readouts) (Dataset EV1). In most cases, the concentration used in our assay either exceeded or was within the range of concentrations reported to cause detectable effects in a variety of cells.

Cells were pre-treated for 1 h with individual CKIs or vehicle (DMSO) and then stimulated for 0.5, 1, 2, and 4 h with LPS or IL-4 and processed for ChIP-seq using a validated antibody for histone H3 acetylated at Lys 27 (H3K27ac). To determine the effects of the CKIs on the epigenome of unstimulated and LPS- or IL-4-stimulated macrophages, we first generated a reference catalog of genomic histone acetylation sites that were centered on the accessible chromatin regions determined by ATAC-seq (assay for transposase-accessible chromatin by sequencing) (Buenrostro et al, 2013). We identified ~16,500 and ~5000 regions displaying dynamic H3K27ac changes after LPS or IL-4 stimulation, respectively (Fig. 1B; Appendix Fig. S2A,B; Dataset EV2). The differential histone acetylation sites could be broadly divided into four LPS- and three IL-4-regulated clusters on the basis of their kinetics (color-coded in Fig. 1B; Appendix Fig. S2A,B).

The effects of the 58 selected CKIs were then determined at the same time points by generating ca. 600 H3K27ac ChIP samples that were sequenced and analyzed in order to measure changes in H3K27ac at each cis-regulatory element (CRE) across all the time points and treatments used (Fig. 1A).

The resulting dataset included a massive array of features, with H3K27ac exhibiting dynamic changes across thousands of CREs over time and in response to treatment with different CKIs.

To analyze such a high-dimensional dataset, a method was required that would fulfill four basic requirements: (i) avoiding any a priori assumptions; (ii) preserving the information on the structure of the data, which is required to disentangle the contribution of individual CKIs to the regulation of each CRE; (iii) reducing data complexity by extracting the most relevant information and the relationships among inhibitors at the level of each individual CRE; (iv) expressing similarities among CKIs as a function of the panel of CREs perturbed, so that the higher the overlap of the epigenomic effects of two inhibitors, the higher their functional similarity.

The method we eventually used was a multi-table dimensionality reduction strategy known as Multiple Factor Analysis (MFA) (Abdi et al, 2013; Escofier and Pagès, 2008). In more detail, the application of MFA facilitates the evaluation of each variable (e.g., H3K27ac, time, CKI used etc.) in relation to the principal source of variation within the dataset by preserving its underlying structure. Schematic representation of the individual steps performed by the MFA algorithm are depicted in Fig. 1C. In our specific case, the ca. 600 H3K27ac ChIP-seq datasets could be structured into 59 groups for both stimuli used (LPS and IL-4), namely one for each of the 58 CKIs in addition to one for the DMSO. Within each group, we cataloged the H3K27ac ChIP-seq read counts corresponding to stimulus-regulated CREs (rows) across time (columns) (Fig. 1C). Eventually, the new set of features generated by the MFA (known as MFA compromise) represents the effects of all different inhibitors

on each LPS-regulated (Fig. 1D) or IL-4-regulated (Appendix Fig. S2B) CRE.

It is crucial to observe that all the H3K27ac ChIP-seq datasets were generated across the same time-course. Hence, the primary components extracted through the MFA were expected to reflect the temporal changes in H3K27ac for each CRE (Fig. S2F,G). This relationship became apparent when we superimposed the CREs onto the first two dimensions of the MFA compromise (Fig. 1D, left), color-coding them according to the clusters of H3K27ac kinetics reported in Fig. 1B and Appendix Fig. S2A,B. Consistent with this observation, CREs located in close proximity within the MFA space demonstrated a greater temporal correlation than randomly selected CREs (Appendix Fig. S2C), thus underscoring the ability of spatial relationships to capture coherent temporal patterns of H3K27ac within the dataset.

Given that the MFA preserves the information of the dataset structure, the contribution of each CKI to the final position of each CRE in the low dimensional space can be extracted. As a representative example, H3K27ac at the CRE upstream of the Brinp3 gene peaked at 2 h post-LPS (Fig. 1D, upper right panel) and its position within the MFA compromise plot primarily reflected such H3K27ac kinetics (Fig. 1D, upper right and left panel). Such position is the barycenter of each CRE and represents the average coordinates computed considering all experimental conditions. By decomposing the H3K27ac data (Fig. 1D, upper middle panel), it was possible to distinguish the impact of each individual CKI on the barycenter of this CRE: while inhibitors such as Mubritinib and Ulixertib only mildly perturbed signal-induced acetylation at this CRE, and thus modestly altered its position in the lower dimensional space relative to that in the DMSO condition, AZD-8186 and Apitolisib contributed to shift the position of this CRE because they caused a complete loss of LPS-induced acetylation (Fig. 1D, bottom right panels). Eventually, the barycenter of this enhancer was shifted away from the DMSO condition because several CKIs impaired its LPS-induced histone acetylation. Hence, the lower-dimensional space defined by the MFA components effectively characterizes both the kinetic mode and the magnitude of perturbation that each CKI exerted on every CRE.

By extracting the distances between the DMSO condition and any CKI for each CRE, within relevant MFA components capturing variance more than randomly permuted data (Fig. 1E; Appendix Fig. S2D), we computed a perturbation likelihood, defined as the probability that a given CKI will perturb a specific CRE (Appendix Fig. S3A). The perturbation likelihood is linearly related to the observed changes in the H3K27ac signal (Fig. 1F; Appendix Fig. S2E). In Fig. 1G we show two representative examples of CREs regulated in an opposite direction by Mubritinib and BGT-226. Overall, H3K27ac was perturbed by at least one CKI at ~85% of the LPS-regulated CREs and at ~89% of the IL-4-controlled CREs (Appendix Fig. S3B). Notably, there were minimal if any differences between affected and unaffected CREs in the amplitude of LPS- or IL-4-induced H3K27ac changes (Appendix Fig. S3C). While most CKIs preferentially caused the downregulation of stimulus-induced H3K27ac, some others preferentially upregulated H3K27ac (Fig. 1H; Dataset EV3). In addition, a few inhibitors caused the nearly selective impairment of H3K27ac changes elicited by either stimulus (Fig. 1H; Dataset EV3).

To analyze the relationships among CKIs, we applied a principal component analysis on the basis of the computed perturbation

likelihood and the direction of change (i.e., up- or down-regulation) per CRE (Appendix Fig. S4A). Twelve significant components, which captured ~70% of the variance, were selected based on iterative permutation testing (Appendix Fig. S4A). In order to visualize such complex multidimensional dataset, we represented the relationships among CKIs with a 2D network-graph in which each vertex represents a CKI that is linked to its three closest CKIs and positioned accordingly (Fig. 1I). Notably, the evaluation of the relative position of individual CKIs showed no evident correlation among CKIs with identical or similar designated target(s). The lack of such correlation is rooted in the poly-pharmacology characteristic of the CKIs. In addition, the relative position and closest neighbors of each CKI were largely different when considering LPS or IL-4 stimulation. A conceivable explanation of this finding is that, due to the inherently low specificity of most CKIs, the effects of individual inhibitors are influenced by the distinct network of signaling pathways regulated by different stimuli (Fig. 1I; Appendix Fig. S4B).

Overall, CKIs with similar designated targets generated largely different effects at the epigenome level, indicating that the scope of the functional effects of CKIs may not be reliably inferred based on the current annotations.

## Similarities among CKIs as a function of kinetics and genomic overlap of H3K27ac changes

Having observed that CKIs with similar designated targets yielded disparate effects at the epigenome level, thus highlighting the limitations of current annotations, we set out to identify the relationships among CKIs captured through the MFA analysis. Our assumption was that the kinetics of CKI-induced changes in stimulus-regulated H3K27ac could offer insights into the signaling pathways and downstream TFs they inhibit. By discerning the temporal trends of H3K27ac changes induced by each CKI, we aimed to infer similarities among these inhibitors. In other words, we hypothesized that if two CKIs lead to H3K27ac changes at an overlapping set of CREs within the same temporal window, this similarity likely stems from the inhibition of common sets of kinases. To explore this hypothesis, we computed the frequency distribution of either up- or down-regulated CREs within the two dimensions of the MFA compromise plot (Fig. 2A), which are informative of the different kinetics of H3K27ac at individual CREs (as shown in Fig. 1B and Appendix Fig. S2A–C). In Fig. 2A, we report three representative examples of the kinetic differences in the effects of individual CKIs. Each CRE, represented as a dot in the 2D space of the MFA compromise, was colored according to the perturbation likelihood score (using opposite color scales for upregulation and downregulation effects), and the frequency distribution of the effects within the four kinetic clusters was assessed (Fig. 2A). While the HER2/ErbB2 inhibitor Mubritinib and the PI3K inhibitor BGT-226 preferentially downregulated H3K27ac at 0.5–1 h post activation, TG-100572 almost exclusively downregulated CREs with H3K27ac peaking at 2–4 h post activation. Although TG-100572 was originally developed as a SRC kinase inhibitor, in kinobeads assays it bound more efficiently to the JAK family member TYK2 (Klaeger et al, 2017), and indeed it was used here at the $EC_{50}$ for TYK2 as measured by kinobeads.

In addition, we assumed that CKIs with identical or similar targets would affect H3K27ac at *overlapping sets* of genomic regions. Therefore, to measure the overlap among the CREs affected by the different CKIs used, we computed the pair-wise Jaccard index to evaluate the extent to which pairs of CKIs perturb H3K27ac at shared groups of CREs (Fig. 2C; Appendix Fig. S4D). Indeed, CKIs identified as similar based on the kinetic analysis in Fig. 1, perturbed common sets of CREs.

Considering insights from our time-based analysis and the assessment of the genomic overlap of CKIs' effects, it is plausible that the identified similarities among CKIs are the consequence of the perturbation of kinases converging on the same signaling pathways. In addition, we observed that the various CKIs exhibited a higher prevalence of downregulated H3K27ac events compared to upregulated events (Fig. 2B,C). Furthermore, a significant proportion of these events were concentrated at CREs induced 2 h post LPS stimulation (Fig. 2B,C and Appendix Fig. S4C,D; Dataset EV3). This observation underscores the predominance of down-regulatory effects across CKIs, as previously described (Fig. 1G), suggesting interference with established signal-transduction mechanisms and their known components. In contrast, despite the prevailing down-regulatory effects observed across CKIs, we discern a notable contribution from up-regulatory events in defining similarities among CKIs (Fig. 2B,C). This phenomenon may be ascribable to the involvement of additional, yet specific, signaling components targeted by groups of CKIs used.

Because of the intrinsic poly-pharmacology of CKIs, it is conceivable that the relationships between CKIs detected with our approach reflected the sum of on- and off-target effects caused by each inhibitor in activated macrophages. To formally address this possibility, we compared the mean distance of inhibitors targeting the same family of kinases ("intra-family") with that of randomly label-shuffled CKIs (Fig. 2D). On average, and in both signaling contexts, distances among CKIs targeting kinases of the same family on the basis of kinobeads-assigned targets, were not only significantly smaller than random but also significantly smaller than the mean distance using the designated targets (Fig. 2D). On the one hand, this observation suggests that kinobeads assays significantly improve specificity profiling of CKIs compared to the annotations reported in clinical labels; on the other, the observation that CKIs targeting completely unrelated kinase families are in some cases equally close to, or even closer than, CKIs targeting kinases in the same family, suggests that the complexity of the observed effects likely arose from CKI-specific combinations of on-target and off-target effects (Fig. 2E).

To further corroborate these observations, we assessed whether CKI proximity within the LPS network could be reflected in similarities in transcriptional profiles. To do so, we generated RNA-seq datasets in primary LPS-stimulated (2 h) macrophages following pre-treatment with individual CKIs or vehicle. The correlation between the changes in gene expression caused by the most proximal CKI pairs was significantly higher than that of randomly sampled pairs (Fig. 2F), indicating the ability of RNA-seq to capture functional similarities among CKIs. Finally, by looking at the top distant pairs of CKIs within the LPS network, the correlation was significantly lower than that of randomly sampled pairs, indicating that the captured H3K27ac similarities effectively encompassed both correlative and anti-correlative transcriptional changes (Fig. 2F; Dataset EV5). In summary, these observations indicate that similarities among CKIs, as inferred from H3K27ac changes, mirror the comprehensive spectrum of transcriptional

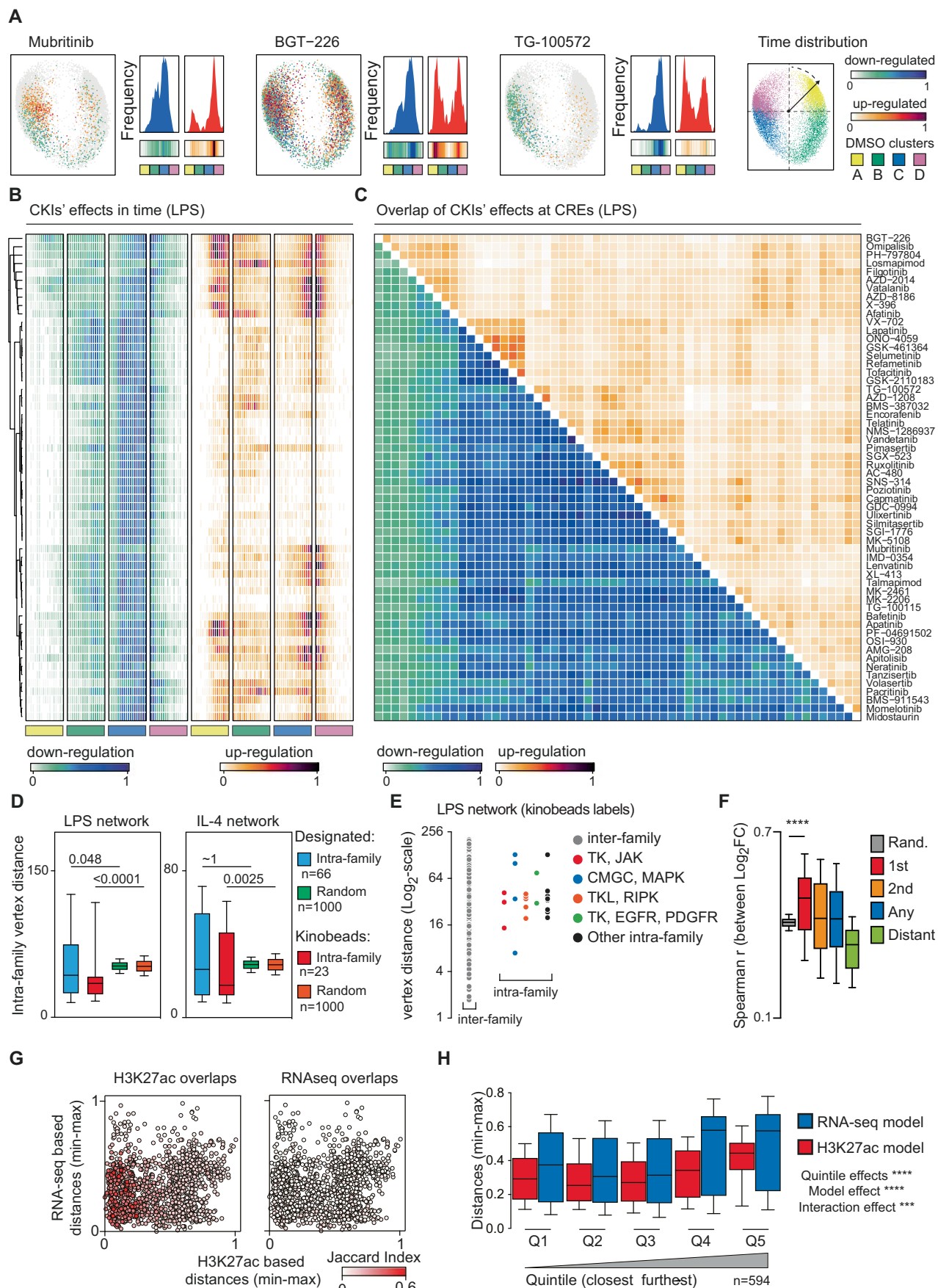

**Figure 2. Similarities among CKIs inferred on the basis of their effects on H3K27ac.**

(A) MFA compromise plots showing the effects on LPS-regulated H3K27ac of three representative CKIs, namely Mubritinib, BGT-226 and TG-100572. For each inhibitor: on the left, the effect of the CKI within the first two dimensions of the MFA compromise is shown (down-regulatory effects on a white-to-blue color scale and up-regulatory effects on a white-to-red color scale); on the right, the frequency distribution of the effects specific to each CKI is displayed, separated into upregulation and downregulation categories. A density heatmap is also provided below, which is color-coded according to the type of regulation. The position of the four kinetic clusters, which were previously defined in Fig. 1B, is shown below each histogram. The frequency was calculated based on the angle formed between the first two dimensions of the MFA, which was determined by dividing the 360° angle into 5° bins and then min-max scaled (0–1). The reference 2D plot can be found on the right, using the same color code as in Fig. 1B, C. (B) Time-resolved effects of CKIs on H3K27ac at LPS-regulated CREs. The heatmap shows the effects of CKIs on H3K27ac at the four main groups of CREs (as shown in Figs. 1B, D) computed as in panel (A). CKIs were clustered based on the computed perturbation likelihood score at each CRE. Cluster branching and naming are identical to panel C. (C) Genomic overlap (standardized effect size, SES, of the Jaccard index, which provides a synthetic description of the extent of the intersection of the CREs affected by two CKIs) of CKIs' effects at LPS-regulated CREs. CKIs were clustered based on the computed perturbation likelihood score at each CRE. Cluster branching is identical to that in panel (B). (D) Mean pair-wise distances between CKIs targeting kinases of the same family ("intra-family") is reported, considering the designated assignment (cyan) or the kinobeads assignment (red). Mean pair-wise distances between CKIs targeting kinases of the same family considering the two assignments of randomly label-shuffled are shown (green box-plot for the designated-assignment and orange for the kinobeads-assignment). The same analysis is reported for both the LPS (left) and the IL-4 (right) computed networks (significance was evaluated with a Kruskal–Wallis test). Box plot showing median, 25th and 75th percentiles (box), 10th and 90th percentile (whiskers), number of pairs for each group are specified within the panel. (E) Pair-wise distances for CKIs targeting kinases of different families (left, "inter-family") or belonging to the same family (right, "intra-family") considering the kinobeads-assignment. (F) Pair-wise Spearman correlation between changes in gene expression, calculated as the RNA-seq $\log_2$ fold-change ($Log_2FC$), comparing individual CKIs to the DMSO condition. Pair-wise mean Spearman r distribution considering the 58 inhibitors and either their closest (red), second closest (orange), any (blue) or the 100 most distant pairs (green) (Kruskal–Wallis test ****$p < 0.0001$). Gray: mean pair-wise Spearman r over 5000 iterations of 58 randomly sampled CKIs. Box plot showing median, 25th and 75th percentiles (box), 10th and 90th percentile (whiskers). (G) Pair-wise distances between CKIs for either model are compared. Each dot represents a CKI pair and reported are the distances between them computed on the basis of the H3K27ac (x-axes) or the RNA-seq (y-axes) at 2 h LPS stimulation. Pairs are colored according to the overlap of the effects (Jaccard Index) detected using the H3K27ac readout (left) or the transcriptional changes (right). (H) Quintile distance distribution to evaluate the model's ability to distinguish the observed effects in the other (Two-way ANOVA test was used to evaluate significance: ***$p = 0.0002$, ****$p < 0.0001$). Box plot showing median, 25th and 75th percentiles (box), 10th and 90th percentile (whiskers). Number of pairs per quintiles are reported in the figure panel ($n = 594$).

alterations induced by each individual inhibitor. However, our initial assumption (and the motivation to use epigenomic rather than transcriptomic data for our study) is that the granularity (i.e., the number of variables) provided by the H3K27ac analysis together with the exclusion of confounding factors such as RNA stability or post-transcriptional regulation, would render H3K27ac a superior read-out for high-resolution analyses of CKI-induced perturbations of signaling pathways. To verify this assumption, we conducted a direct pair-wise comparison of CKI proximity based on either H3K27ac alterations or transcriptional changes detected by RNA-seq. This comparison focused solely on the 2-h stimulation time point in both datasets, following the same analytical framework outlined above and resulted in the generation of H3K27ac and RNA-seq-based models of CKI proximity (Fig. 2G,H). Pair-wise distances between CKIs using both models were compared and related to the degree of overlaps between CKIs of the pair (Jaccard index, Fig. 2G). Compared to the RNA-seq computed model, H3K27ac ChIP-seq was capable of capturing a higher degree of overlaps between CKIs. Furthermore, when we sorted all CKI pairs into quintiles based on distances calculated using the H3K27ac data, we achieved a more pronounced separation between closely related CKIs and distantly related CKIs compared to their distances in RNA-seq data (Fig. 2H).

Taken together, these findings suggest that H3K27ac changes can capture similarities among CKIs with higher resolution than transcriptional data.

## Deciphering the mechanistic impact of CKIs on the epigenome through TF regulation

To unravel the intricate epigenomic effects induced by a diverse set of CKIs, we employed a comprehensive approach centered on the identification of transcription factors (TFs) regulated by each CKI. Our primary objective was to elucidate how CKIs influence the

epigenomic landscape by modulating the activity of specific TFs. To assess the selectivity of the 58 CKIs used on TF activity, we investigated the association between CKI-induced changes in H3K27ac and TF binding to CREs based on *ca*. 200 high-quality ChIP-seq datasets for 34 TFs in various stimulation conditions (Fig. 3A; Dataset EV4). Consistent with the fact that the CKIs were selected based on their potential to inhibit components of the LPS or IL-4 signaling pathways, our analysis revealed that several TFs were significantly associated with CREs displaying reduced H3K27ac in response to individual inhibitors (Fig. 3A). In contrast, the association of the CKIs' up-regulatory effects with TFs was more heterogeneous, with only a few TFs being enriched and the majority being significantly underrepresented (Fig. 3A). The most parsimonious explanation of this finding is that additional TFs not present in the datasets used may be selectively associated with increased acetylation.

As the effects caused by each CKI depend on the combination of all the signaling pathways and downstream TFs affected, such effects should in principle be predictable using machine learning approaches using as features TF genomic occupancy and H3K27ac kinetic data.

To this aim, we first extracted a set of non-redundant features able to explain the variance of the system. These features were derived by combining the information on signal intensity from approximately 200 TF ChIP-seq experiments (Fig. 3B, top), the MACS2 called ChIP-seq peaks (Zhang et al, 2008) (Fig. 3B, center), and the normalized read counts of H3K27ac per CRE following either LPS or IFNβ stimulation (Fig. 3B, bottom). The resulting 21 features we extracted (Appendix Fig. S5A) proved adequate in explaining ~90% of the overall variance associated with the ca. 16,500 LPS-regulated CREs (Fig. 3B). Notably, the initial three components, explaining ~40–45% of the total variance, corresponded to: (i) activation of signal-induced TFs, as this dimension effectively distinguished between treated and untreated

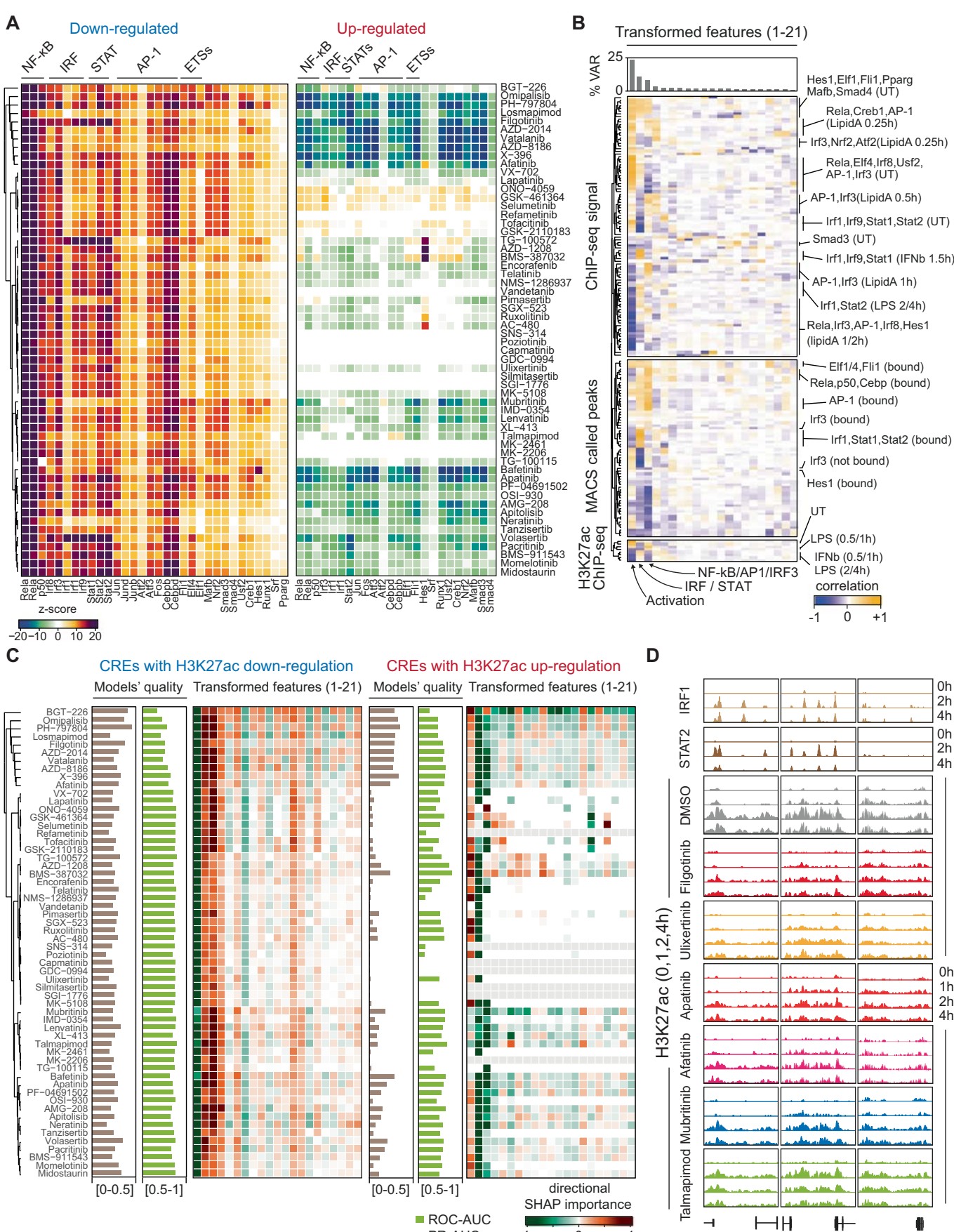

◀ **Figure 3. Linking CKIs' effects to the regulation of distinct sets of transcription factors.**

(A) Association analysis between CKIs-induced H3K27ac down- (left) or up-regulation (right) and TF ChIP-seq peaks. 32 out of 43 datasets are shown considering peaks called in any condition; iterative resampling bootstrap tests were computed using the permTest function of the regionR Bioconductor library. (B) Correlation between original and transformed features (applying dimensionality reduction) exploited to assemble the individual models. The percentage of explained variance by components (top) is reported together with the correlation between transformed features and the signal intensity detected in the ~200 ChIP-seq, MACS2 called ChIP-seq peaks and the H3K27ac signal following LPS or IFNβ stimulation. (C) Summary of the 58 classification models reporting ROC-AUC, PR-AUC and the directional SHAP values (computed as the average SHAP multiplied by the correlation sign) by individual classes (i.e., downregulation on the left and upregulation on the right). (D) Representative examples of H3K27ac changes at the *Mx1*, *Rsad2*, and *Nfkbia* loci following the indicated treatments. ChIP-seq tracks of IRF1 and STAT2 are shown at the top.

TF ChIP-seq samples (Fig. 3B and Appendix Fig. S5B); (ii) activation of IRF-STAT families TFs and IFNβ-dependent H3K27ac changes (Fig. 3B; Appendix Fig. S5C); (iii) binding of NF-kB, AP-1, and IRF3, along with the early induction of H3K27ac upon LPS stimulation (Fig. 3B; Appendix Fig. S5D).

Next, to predict CKI-specific H3K27ac perturbations, we trained 58 independent supervised classification models (Chen and Guestrin, 2016) (Fig. 3C). Each model was tailored to each CKI, and its performance in predicting H3K27ac changes was evaluated by assessing the area under the curve (AUC) for both the receiver operating characteristic (ROC) and precision-recall (PR) curves, considering either down- and up-regulated CREs (Fig. 3C). Based on both measures, models performed better at predicting down-regulatory than up-regulatory effects. Only a few models of adequate quality were instead associated with up-regulatory effects due to the limited number of upregulated CREs specific to individual CKIs (Fig. 3C). A strength of the approach used is that it returns the relative importance of individual features in the prediction. Analysis of feature importance (Lundberg and Lee, 2017) demonstrated that features associated with both IRF/STAT (second component) and NF-kB, IRF3, and AP-1 activation (third component) collectively exerted a positive influence on predicting down-regulatory effects (Fig. 3C, left). Notably, the relative importance of these two features varied among CKIs, aligning with their hierarchical clustering. Specifically, for certain inhibitors (e.g., Filgotinib), the IRF/STAT feature set had a more significant impact on predicting downregulation, while for others (e.g., Mubritinib), the NF-kB/AP-1/IRF3 feature set dominated. These observations were partly corroborated by examining H3K27ac changes at IRF/STAT-bound loci (Fig. 3D). To visually illustrate the models' predictive power, decision plots for two exemplary cases, Filgotinib and Midostaurin, are reported in Appendix Fig. S6A–D. These plots provide a detailed view of the model's decision for each observation, with each observation corresponding to one of the 16,000 CREs in our scenario. Notably, both models effectively segregated CREs into upregulated or downregulated categories. Moreover, the super-imposition of the ChIP-seq signal of selected TFs, such as STAT2, REL/NF-kB, and IRF3, on the individual CREs, revealed that CREs downregulated by Filgotinib exhibited enhanced STAT2 binding. Opposite effects were observed for CREs upregulated by Filgotinib (Appendix Fig. S6A,B). Conversely, Midostaurin-induced down-regulatory effects were associated with IRF3 and REL/NF-kB binding, while up-regulatory effects displayed a complementary pattern.

Predicting up-regulatory effects using TF ChIP-seq data as training features presented a notable challenge in our analysis primarily because of the sparsity of up-regulatory effects, which constituted a minor fraction of the overall effects for the majority of

the CKIs examined. Nonetheless, within models with acceptable quality, the first feature (corresponding to the first extracted component), which is indicative of general TF activation by stimulation (as illustrated in Fig. 3B and Appendix Fig. S5B), had an adverse impact on the prediction of up-regulatory events. When considering up-regulatory effects, we can envisage two plausible and not mutually exclusive scenarios: (a) enhancement of induction by CKIs: stimulus-induced CREs may display greater induction when macrophages are treated with specific inhibitors; (b) mitigation of downregulation: some CKIs may negatively impact part of the LPS-induced H3K27ac downregulation events. Both scenarios entail a higher mean signal across time in the inhibitor condition compared to the DMSO condition. Our analysis revealed that the majority of the models capable of predicting up-regulatory effects were associated with the second scenario. Only a selected subset of inhibitors, specifically Mubritinib, IMD-0354, Lenvatinib, XL-413, Talmapimoid, and Bafetinib, exhibited up-regulatory effects that could be reliably predicted based on signal-induced H3K27ac changes (Fig. 3C, right).

Altogether, these results suggest that genome-wide TF binding patterns can aid in predicting the epigenomic effects of individual CKIs. However, it is important to notice that we cannot fully disentangle which combination of inhibited kinases leads to a set of H3K27ac changes. As a result, we can only correlate H3K27ac alterations with sets of TFs but not directly link specific kinases to TFs. Nevertheless, H3K27ac changes driven by individual CKIs could be related to specific families of TFs.

## Distinctive and unique properties of individual JAK inhibitors (JAKi)

As a specific test case, we analyzed similarities among CKIs targeting the same family of kinases. Given their therapeutic relevance for autoimmune and inflammatory disorders we focused on the CKIs targeting JAK family kinases (O'Shea et al, 2015; Philips et al, 2022; Zarrin et al, 2021).

Out of the 58 CKIs used in this study, 6 were developed to target members of the JAK family, which includes four kinases (TYK2, JAK1, JAK2, and JAK3) involved in IFN and cytokine signaling and implicated in the pathogenesis of autoimmune and inflammatory disorders (Philips et al, 2022) (Fig. 4A; Dataset EV1). The designated specificities of these six CKIs, based on FDA approvals or reported in clinical trials, included individual JAKs or their combination and in some cases additional kinases (e.g., Pacritinib was approved as both a JAK2 and a FLT-3 inhibitor) (Dataset EV1). However, specificities identified in kinobeads data (Klaeger et al, 2017) differed extensively from those indicated in official clinical approvals. Notably, in kinobeads assays Filgotinib, Tofacitinib, and

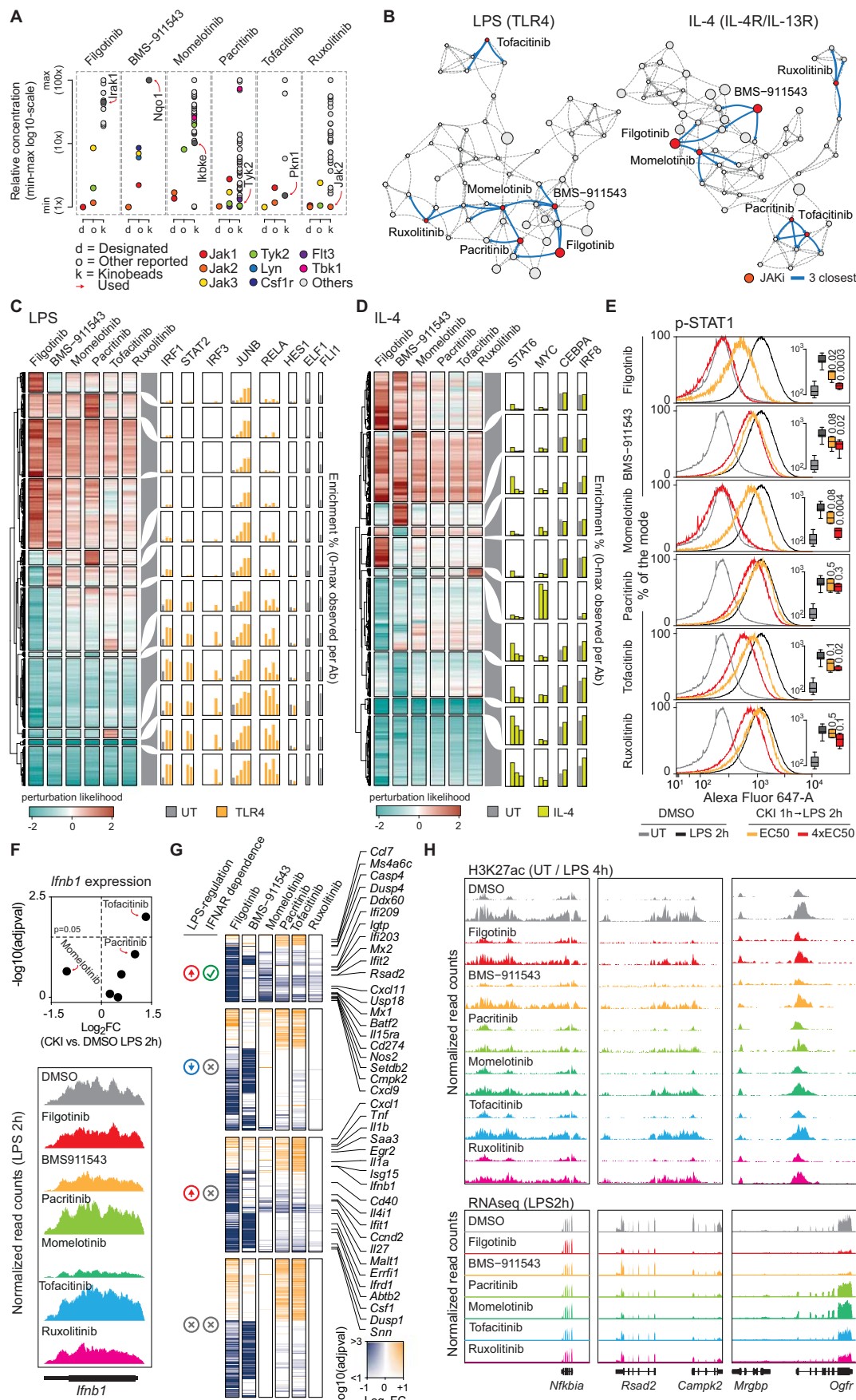

◄ **Figure 4.  Dissection of similarities and divergences among JAK family kinase inhibitors.**

(A) Relative concentration scheme reporting min–max values on a $\text{Log}_{10}$ scale. For each CKI, concentrations for the designated target (d), the targets measured in kinobeads assays (k) and other reported targets (o) are shown. The concentration used is indicated by a red arrow. (B) JAKi positions within the LPS and IL-4 networks are highlighted in red together with the three closest connections (blue lines). (C, D) CREs affected by any JAKi (perturbation likelihood > 1) clustered by the perturbation likelihood for the 6 JAKi in LPS (C) or IL-4 (D), together with the relative abundance of a selection of TFs ChIP-seq. (E) Histogram reporting the percentage of the mode relative to the p-STAT1 signal (Alexa Fluor 647-A) in macrophages pre-treated with DMSO or with the individual CKIs as indicated, before or after LPS stimulation (2 h). Concentrations corresponding to the $\text{EC}_{50}$ for the intended target or $4\text{x}\text{EC}_{50}$ were used, as indicated. Inside individual histograms, box plots are shown reporting the mean p-STAT1 signal measured by FACS in triplicate experiments. Box plot showing median, 25th and 75th percentiles (box), 10th and 90th percentile (whiskers). (F) Top: Scatter plot of *Ifnb1* gene expression reporting the $\text{Log}_2\text{FC}$ (x-axes) and the $-\text{Log}_{10}$(adjpval, computed as for DESeq2, i.e., Wald test of the negative binomial model coefficients) (y-axes) for each JAKi. Bottom, snapshot of the *Ifnb1* gene for the DMSO condition and the indicated JAKi. (G) Summary heatmap of the RNA-seq results reporting any differentially expressed genes compared to the DMSO condition and any JAKi. The list of genes is divided into 4 groups according to LPS regulation (increase or decrease) and IFNAR dependency. $\text{Log}_2\text{FC}$ changes are reported on a blue-to-orange color scale, with the opacity scaled according to the statistical significance computed by DESeq2 (Wald test of the negative binomial model coefficients). (H) Representative examples showing both H3K27ac ChIP-seq and RNA-seq signals at the *Nfkbia*, *Rsad2*, and *Ogfr* loci.

BMS-911543 were not found to be associated with JAKs and they were instead found to target other kinases (Dataset EV1). Figure 4A reports the concentrations used in our screening and their additional targets based on kinobeads data.

We examined the relative position of these 6 JAKi in both the LPS and the IL-4 2D network-graph, in which each vertex represents a CKI that is linked to the three CKIs causing the most similar epigenomic effects, and positioned accordingly (Fig. 4B). In both signaling contexts, the six JAKi did not cluster together and their closest neighbors were often unrelated CKIs. The only two exceptions were represented by BMS-911543, which was directly linked to Filgotinib and Momelotinib in the LPS network, with BMS-911543 being the closest CKI to Filgotinib and Momelotinib the closest to BMS-911543 (Fig. 4B).

Downregulated CREs perturbed by all JAKi showed very limited variability and they were extensively bound by IRF and STAT family TFs in both the LPS and the IL-4 network (Fig. 4C,D), confirming that these inhibitors, albeit with varying potency, are all capable of reducing the activity of JAK kinases. In contrast with the homogeneity of down-regulatory effects, a large variability was instead found when considering CREs showing increased H3K27ac, and such differences could be ascribed to specific TFs (Fig. 4C,D). For instance, in response to LPS Ruxolitinib mildly upregulated H3K27ac at CREs bound by JUNB (Fig. 4C). In the context of IL-4 stimulation, it strongly and selectively upregulated H3K27ac at enhancers bound by MYC, which were instead downregulated by Filgotinib, BMS-911543 and Momelotinib (Fig. 4D).

To directly evaluate the effects of the six JAKi used, we performed a flow cytometry assay to quantitatively measure JAK-mediated phosphorylation of STAT1 at Tyr701 (p-STAT1) after LPS stimulation, in the absence or presence of each individual inhibitor (Fig. 4E). Notably, at the concentration used in our genome-wide screen (Dataset EV1) only Filgotinib and Momelotinib caused a clear and significant reduction in p-STAT1, while the others caused minimal and/or no significant changes. It is important to note that the concentration employed in the genome-wide screen and in this flow cytometry experiment corresponds to the $\text{EC}_{50}$ determined by kinobeads assays, which provide an indirect measure of compound activity that may not be equivalent to the $\text{EC}_{50}$ values determined in cell-based assays. At a 4-fold higher concentration ($4\text{x}\text{EC}_{50}$), we observed inhibitory effects for all these inhibitors, with Filgotinib and Momelotinib completely abrogating STAT1 phosphorylation (Fig. 4E). Overall,

while Filgotinib and Momelotinib caused quantitatively similar effects on p-STAT1, their relative positions in the 2D network graphs (Fig. 4B) as well as the complement of their effects on H3K27ac at CREs (Fig. 4C,D) were substantially different. Importantly, when considering genome-wide H3K27ac changes, which clearly exceed the sensitivity of p-STAT1 detection by flow cytometry, we consistently observed that all JAK inhibitors impaired a common set of CREs associated with IRF/STAT binding. Hence, the discrepancy in their relative positions and the differential impact on up-regulatory effects suggest that on-target effects accounted for only a fraction of the broader cellular effects induced by these inhibitors.

Impairment of the JAK-STAT signaling pathways in LPS-stimulated macrophages could be a consequence of the direct inhibition of JAK activation downstream of the IFNAR and/or be due to reduced *Ifnb1* gene induction, and then IFNB1 secretion, *via* TBK1-mediated phosphorylation of IRF3 (Sharma et al, 2003). Analysis of the RNA-seq datasets in LPS-stimulated cells showed that among the six JAKi used, Momelotinib was the only one able to reduce *Ifnb1* gene expression (Fig. 4F), suggesting that its effects on STAT1 phosphorylation (Fig. 4E) could be due to both direct effects on JAKs and to reduced *Ifnb1* expression caused by TBK1 inhibition. This possibility is consistent with previous data (Tyner et al, 2010) and partially with kinobeads measurements, in which Momelotinib was found to bind also to TBK1, albeit at a 2.5-fold higher concentration than that we used here.

Inspection of the RNA-seq data revealed that, despite their distinct potencies, all JAK inhibitors influenced IFN-regulated genes, consistent with their intended mechanism of action. However, these IFN-regulated genes constituted a minority of the differentially expressed genes (DEGs), and the overlaps of the effects of the JAK inhibitors were limited, a pattern in line with the genome-wide H3K27ac changes. Consistent with the p-STAT1 data (Fig. 4E), Filgotinib was the JAKi causing the greatest impairment of IFN-stimulated genes (Fig. 4G). Nevertheless, coherently with the 2D network based on H3K27ac changes (Fig. 4B), the six JAKi displayed little correlation at the transcriptional level, with the exception of BMS-911543 and Filgotinib, which shared an extensive set of differentially regulated genes, most of them being, however, not regulated by IFNβ (Fig. 4G; Appendix Fig. S7A). H3K27ac and RNA-seq data for representative genes are shown in Fig. 4H, with *Nfkbia* representing a canonical NF-kB-regulated gene, *Rsad2* and *Cmpk2* two IFN-stimulated genes and *Ogfr* an LPS-induced but IFN-independent gene.

Overall, these data indicate that each JAKi induced highly specific epigenomic and transcriptional effects. Therefore, every JAKi is a unique molecule with a distinct spectrum of regulatory effects. Moreover, off-target effects played a prominent role in shaping the observed regulatory changes even though the JAK inhibitors were used at lower concentrations than those required to fully inhibit their intended targets. Hence, the high granularity of genome-wide H3K27ac changes allowed us to identify both JAK-dependent targets affected by all inhibitors and a multitude of off-targets specific to each inhibitor, again highlighting the poly-pharmacological nature of these compounds as the main confounder of the measured effects.

## Unexpected effects of CKIs on the IFNβ-regulated gene expression program

*Ifnb1* gene induction is the driver of a large share of the secondary effects caused by LPS stimulation of macrophages and is controlled by a complex and highly orchestrated combination of TFs. The observation that *Ifnb1* gene induction was impaired by Momelotinib prompted us to assess whether any other CKI could impact its regulation (Fig. 5A). Unexpectedly, several inhibitors led to the upregulation of *Ifnb1*, including the PI3K inhibitors Omipalisib and BGT-226, the HER2/ErbB2 inhibitors Mubritinib and Lapatinib, the IKBKB inhibitor IMD-0354, the multi-kinase inhibitor Lenvatinib and the JAK inhibitor Tofacitinib (Fig. 5A). Notably, these CKIs did not form a coherent cluster in the LPS and the IL-4 network graphs, indicating otherwise largely divergent epigenomic effects (Fig. 5B). We first evaluated whether the increased *Ifnb1* gene expression caused by these CKIs was due to the increased activity of the TBK1/IRF3 pathway. To this aim we determined if the CREs showing H3K27ac gain in response to treatment with these CKIs were bound by IRF3 (Fig. 5C). IRF3 was found to bind CREs at which H3K27ac was induced by BGT-226, Lapatinib and Tofacitinib (Fig. 5C), suggesting that the increased induction of the *Ifnb1* gene by these CKIs may relate to TBK1 hyper-activation. Moreover, all these three CKIs caused an increase in H3K27ac in basal (non-LPS-stimulated) macrophages at CREs bound by both ATF2 and JUN. Notably, based on RNA-seq data, all these three inhibitors caused pervasive upregulation of gene expression that also included the super-induction of IFNβ-regulated genes (Fig. 5D) and other groups of immune response genes (Fig. 5E).

We next attempted to further rationalize the mechanistic bases of the observed hyper-induction of *Ifnb1*. A common mechanism underlying *Ifnb1* gene induction is the viral mimicry caused by the activation of repeat elements such as the ERVs (endogenous retroviruses), followed by the accumulation in the cytoplasm of nucleic acids (such as double-stranded RNAs) whose sensing by pattern recognition receptors triggers the activation of the TBK1-IRF3 pathway (Chen and Hur, 2022). Notably, the MEK1/2 inhibitor Trametinib was shown to be able to induce retroelement activation and *IFNB1* gene expression in pancreatic cancer cells (Cortesi et al, 2024). Therefore, we explored whether any of the seven CKIs increasing *Ifnb1* expression had an effect on the transcription of transposable elements (TEs). We used our RNA-seq dataset to determine both locus-specific and family-wise induction of TE expression. Treatment of mouse macrophages with any of the seven inhibitors resulted in the induction of numerous TEs (Fig. 5F) and in particular ERVs, which can generate

double-stranded RNAs and thus activate the IFN response (Fig. 5F). While this analysis does not pinpoint the specific mediators responsible for the upregulation of the repeat elements, it suggests that deregulated expression may underlie *Ifnb1* induction in response to specific CKIs and highlights the potential for our approach to identify off-target effects that may give rise to unexpected phenotypic outcomes.

In addition to the CKIs leading to the upregulation of the *Ifnb1* gene, Midostaurin (commercially known as Rydapt or Tauritmo) and to a lesser extent Momelotinib, both had a negative impact on *Infb1* induction (Fig. 5A). Inspection of the 2D LPS network graph highlighted that Momelotinib was indeed the closest neighbor of Midostaurin (Fig. 6A, left). This link was instead lost when considering the IL-4 network graph in which Momelotinib was placed further away from Midostaurin and instead close to Filgotinib (Fig. 6A, right).

Midostaurin is a multi-kinase inhibitor approved for the treatment of systemic mastocytosis and acute myeloid leukemia with FLT-3 activating mutations (Kim, 2017). Kinobeads data indicate that TBK1 was among the kinases with the highest affinity for Midostaurin, and indeed in our screening Midostaurin was used at the $EC_{50}$ for TBK1 (Fig. 6B). On the other hand, Momelotinib, a recently approved multi-kinase inhibitor for the treatment of myelofibrosis and anemia, was used in this study at a concentration targeting IKBKE, a TBK1 paralogue (Fig. 6B). Consistent with the proximity between Midostaurin and Momelotinib in the 2D network graph, their effects on H3K27ac were extremely similar and no obvious specificities for subsets of CREs were observed (Fig. 6C). Moreover, also their effects on gene expression measured by RNA-seq were similar (Fig. 6D; Appendix Fig S7B). Consistent with a potential inhibition of TBK1, a large number of IFN-dependent genes was impaired by both Midostaurin and Momelotinib (Fig. 6D). Moreover, IRF3 binding to the CREs down-regulated by both inhibitors was highly significant (Fig. 6E), phosphorylation of STAT1 by LPS was reduced (Fig. 6F) and expression of *Ifnb1* as well as IFN-stimulated genes was almost abrogated (Fig. 6G).

To directly assess the interaction between Midostaurin and Momelotinib with TBK1 in living cells, we performed cellular thermal shift assay (CETSA) (Martinez Molina et al, 2013) in macrophages pre-treated with Midostaurin or Momelotinib and then stimulated with LPS for 2 h. This assay is based on the biophysical principle of ligand-induced thermal stabilization of a target protein, which in principle enables the direct measurement of the engagement of cellular targets in whole cells, although effects are often complex and of difficult interpretation (Seashore-Ludlow et al, 2020). Midostaurin was able to increase TBK1 thermal stability at the $EC_{50}$ for TBK1 (Fig. 6H), while Momelotinib had a limited if any impact at the $EC_{50}$ for IKBKE or at 4x this concentration, while causing an increase in TBK1 thermal stability at higher concentration ($10xEC_{50}$ for IKBKE) (Fig. 6I). This observation was consistent with kinobeads data in which Momelotinib was able to bind TBK1 at concentrations higher than those found to affect its paralog IKBKE. This observation suggests that the genome-wide strategy we established was capable of capturing relationships among CKIs that may be only partially detectable or completely undetectable using thermal stability assays. In addition, the similarities of the epigenomic effects of these two compounds when used at the $EC_{50}$ for TBK1 and IKBKE, respectively, occurred

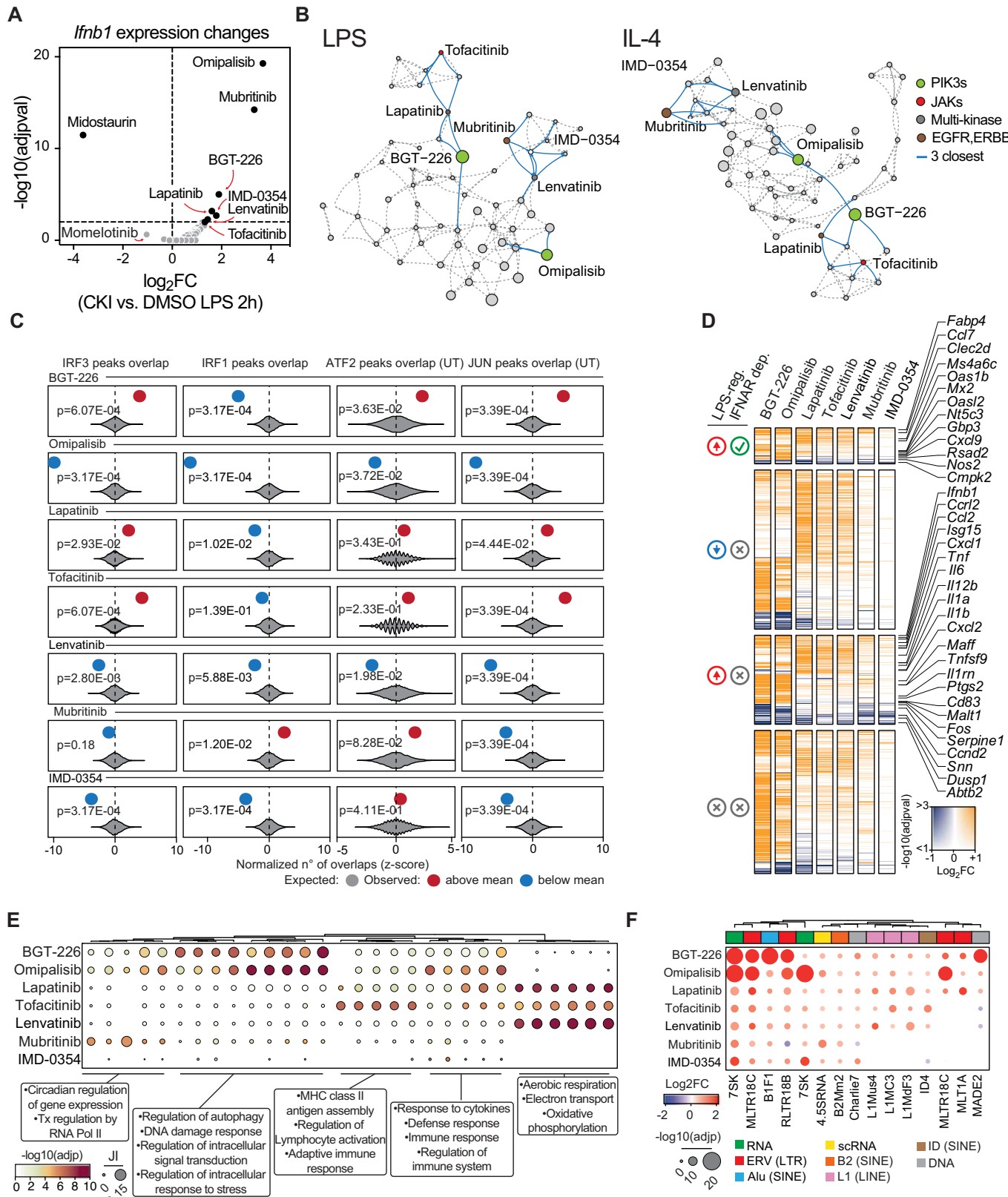

**Figure 5.  Mechanistic deconvolution of CKIs enhancing Ifnb1 gene expression.**

(A) Volcano plot of *Ifnb1* gene expression as in Fig. 4F, reporting *Ifnb1* Log$_2$FC (x-axis) and −Log$_{10}$(adjpval, computed as for DESeq2, i.e Wald test of the negative binomial model coefficients) for all the 58 inhibitors used. (B) The positions within the LPS and IL-4 networks of any CKI enhancing *Ifnb1* gene expression are highlighted together with the three closest connections (blue lines). (C) Bootstrap randomization test reporting the expected normalized number of overlaps by random (gray violin) between sampled CREs and IRF3, IRF1, ATF2, and JUN peaks. The observed normalized overlaps (z-scored) are shown in red (when above) and blue (when below) the mean by random. Reported is the multiple-test corrected *p*-value computed via Bootstrap (*n* = 5000) randomization testing and corrected via Benjamini & Hochberg method. (D) Summary heatmap of the RNA-seq results reporting any differentially expressed genes (compared to the DMSO condition) divided from top to bottom into: LPS-induced and IFNAR-dependent genes; LPS downregulated genes; LPS-induced and IFNAR-independent genes; non-LPS-controlled genes. Log$_2$FC are reported on a blue-to-orange color scale, with the opacity scaled according to the statistical significance computed by DESeq2. (E) Gene ontology categories significantly enriched (adjp ≤ 0.01) in genes upregulated by the indicated inhibitors. Dots are color-coded according to the −log10(adjpval) and their size is proportional to the Jaccard Index between MSigDB gene sets and the individual upregulated gene set. (F) Analysis of TE elements (aggregated by family) significantly upregulated following the treatment with each reported inhibitor. Dots are color-coded by the DEseq2 computed Log$_2$Fold change and the size is proportional to the DEseq2 computed adjusted *p*-value (Wald test of the negative binomial model coefficients).

in the absence of detectable alterations in the activation of the other main LPS-activated kinases (Appendix Fig. S7C–F).

To extend our findings, we investigated the impact of Midostaurin, Filgotinib, and Momelotinib treatments on human cells. We used the THP1 human monocyte cell line, which can be induced to differentiate into macrophage-like cells (Nascimento et al, 2022). We assessed the expression of the *IFNB1* and *RSAD2* genes, indicative of TBK1-IRF3 and JAKs-STATs signal transduction and activation, respectively, in THP1 cells treated with LPS, IFNβ or double-stranded DNA (dsDNA), which similarly to LPS induces IFNB1 *via* TBK1 (Tan et al, 2018). Consistent with observations in mouse macrophages, Midostaurin significantly reduced *IFNB1* gene expression after both LPS and dsDNA stimulation, whereas Momelotinib exhibited a weaker effect (Fig. S7G). Conversely, consistent with its specificity for JAK kinases and lack of activity on TBK1, Filgotinib did not reduce *IFNB1* gene expression at any concentration tested and regardless of the stimuli used. Furthermore, while Midostaurin, Filgotinib, and Momelotinib all impacted the dsDNA-dependent induction of the IFN-stimulated gene *RSAD2*, only the latter two compounds reduced it expression upon IFNβ stimulation as a consequence of JAK inhibition (Fig. S7H).

Overall, these data indicate that some CKIs with completely different affinity profiles in kinobeads may nevertheless have extensively similar epigenomic effects and therefore may regulate shared pathways of potential practical and therapeutic relevance.

## Discussion

Poly-pharmacology is a common and well-known property of most small molecules, including KIs approved or tested for clinical use (Ayala-Aguilera et al, 2022; Roskoski, 2022). Based on kinobeads assays, clinically approved KIs were found to possess specificity profiles comparable to those of molecules in early preclinical development (Klaeger et al, 2017). In particular, low specificity does not preclude clinical approval, as exemplified by Midostaurin, a broad-spectrum CKI that is approved for treatment of FLT3$^+$ acute myelogenous leukemias (Levis, 2017). On the other hand, lack of detailed profiling and understanding of cellular and organismal effects of CKIs (as well as any other small molecules) contributes to their high attrition rates, approval failures, and undesired effects in patients.

Initial preclinical validation approaches for small molecules can be broadly grouped into genetic tests and activity studies. In the

former, the effects of a drug on a defined readout (such as cancer cell growth or survival) is measured in the presence or absence of the intended target protein or mutation (Bhattacharjee et al, 2023; Lin et al, 2019). This kind of approach provides a stringent validation of the requirement for the intended target in the effects of a small molecule and is thus critical to define the mechanism of action of the drug. However, it does not provide a comprehensive view of the cellular effects of the small molecule, which are determined by variable combinations of on-target and off-target activities that eventually impact both clinical efficacy and toxicity.

Biochemical activity studies, including phosphoproteomics, complement genetic tests in that they can be used both to measure on-target effects of a drug (e.g., inhibition of a given enzymatic activity) and to more comprehensively characterize the consequences of cellular or organismal exposure to the small molecule under investigation. Nevertheless, despite advancements in phosphoproteomics techniques (Lee et al, 2024), predicting the scope of functional effects of kinase inhibitors in cells still remains a challenge.

CKI targets reported in current clinical annotations are often extensively discordant from those identified by in vitro binding assays, notably kinobeads, which can measure CKI affinities towards a large, yet partial fraction of the kinome. A simple assumption is that by providing an extensive annotation of binding specificities of individual CKIs, in vitro binding data should lead to a more accurate anticipation of the cellular effects of CKI treatments.

In order to determine how targets of CKIs identified by in vitro binding assays relate to cellular effects, we have used a novel strategy grounded on the in-depth deconvolution of unbiased and high-content in vivo generated datasets reporting the measurement of a chromatin modification rapidly responsive to upstream signaling. By applying this approach to a dynamic system in which a large number of signaling events is controlled by acute stimulation, we could obtain a high-resolution analysis of the effects of CKIs with reported overlapping or distinct specificities. As compared to methods based on gene expression profiling, an epigenome-centered approach provides a much higher level of granularity and resolution. Indeed, the measurement of a large number of individual elements (promoters and enhancers, which largely exceed the number of genes) in dynamic conditions, combined with knowledge of the basic grammar of *cis*-regulatory elements (namely TF binding motifs), enabled extensive data analysis, including machine-learning approaches to identify similarities among CKIs also based on the terminal effectors

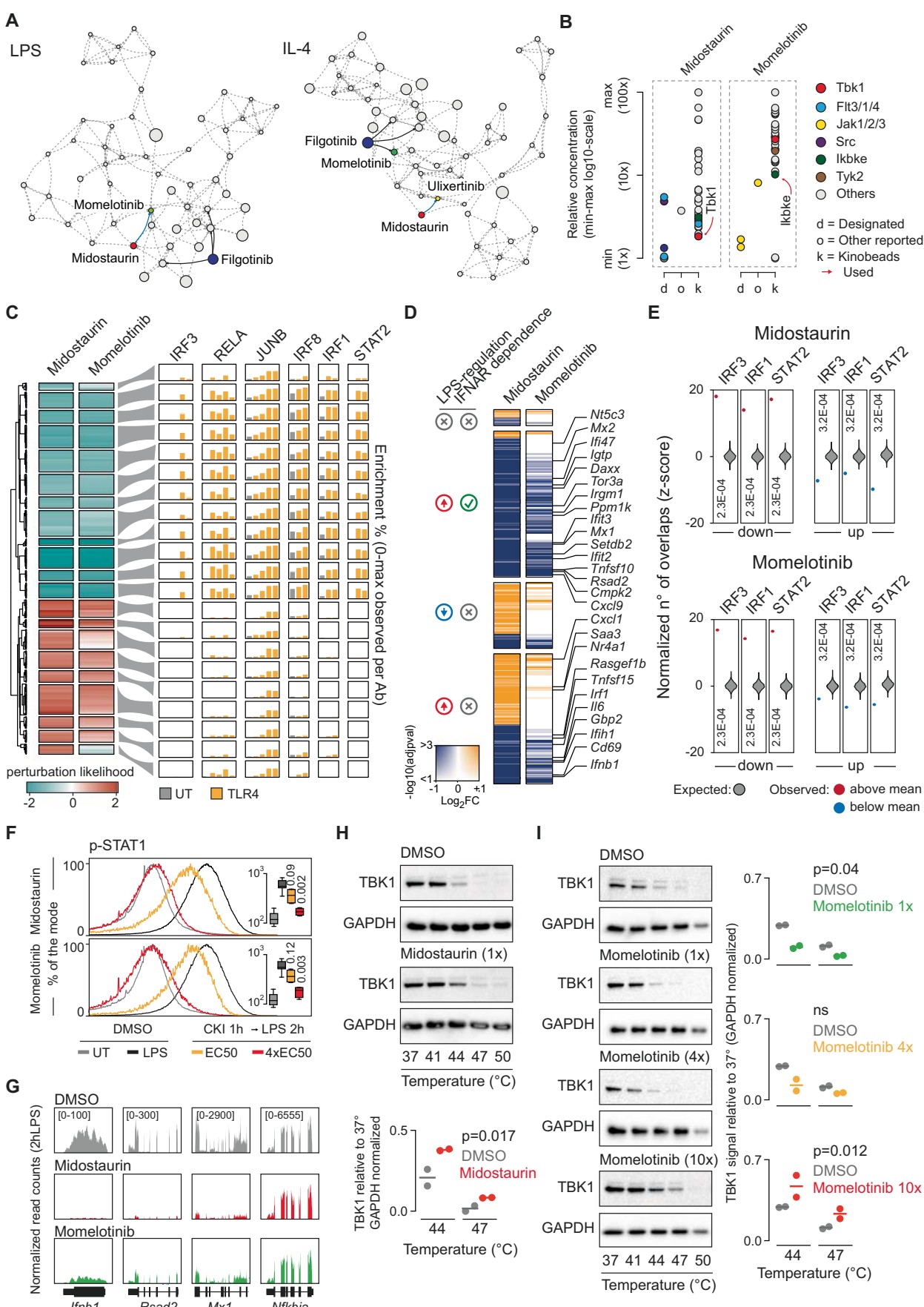

**Figure 6.  Mechanistic deconvolution of CKIs reducing Ifnb1 gene expression.**

(A) Positions within the LPS and IL-4 networks of CKIs reducing *Ifnb1* expression together with the JAKi Filgotinib. The closest connection to Midostaurin is shown with a blue line, while the three closest connections to Filgotinib are shown as black lines. (B) Relative concentration scheme reporting min–max on a $\log_{10}$ scale. For each inhibitor concentrations for the designated target (d), the targets identified by kinobeads assays (k) and other reported targets (o) are shown. The concentration used is indicated by a red arrow. (C) CREs affected by either Midostaurin or Momelotinib (perturbation likelihood > 1) clustered by the perturbation likelihood for both Midostaurin and Momelotinib in LPS (left), together with the relative abundance of a selection of TFs ChIP-seq (right). (D) Summary heatmap of the RNA-seq results reporting any differentially expressed genes (compared to the DMSO condition), divided from top to bottom into: LPS-induced and IFNAR-dependent genes; LPS downregulated genes; LPS-induced and IFNAR-independent genes; non-LPS-controlled genes. $\log_2FC$ are reported on a blue-to-orange color scale, with the opacity scaled according to the statistical significance computed via DESeq2 (Wald test of the negative binomial model coefficients). (E) Bootstrap randomization test reporting the expected normalized number of overlaps by random (gray violin) between sampled CREs and IRF3, IRF1 and STAT2 peaks. The observed normalized overlaps between peaks of any of the three TFs and CREs down- or up-regulated by either Midostaurin (top) or Momelotinib (bottom) are shown in red when above and blue when below the mean by random. Reported is the multiple-test corrected p-value computed via Bootstrap ($n = 5000$) randomization testing and corrected via Benjamini & Hochberg method. (F) Histogram reporting the percentage of the mode relative to the p-STAT1 signal (Alexa Fluor 647-A) in macrophages pre-treated with DMSO or with the individual CKIs as indicated, before or after LPS stimulation (2 h). Concentrations corresponding to the $EC_{50}$ for the intended target or $4xEC_{50}$ were used, as indicated. Inside individual histograms box-plots are shown reporting the mean p-STAT1 signal measured by FACS in triplicate experiments. Box-plot showing median, 25th and 75th percentiles (box), 10th and 90th percentile (whiskers). (G) Representative examples reporting RNA-seq data at the *Infb1, Rsad2, Mx1* and *Nfkbia* genes loci. (H) Representative western blots for the CETSA thermal stability assays for both GAPDH (control) and TBK1 in macrophages treated with DMSO or Midostaurin (top). The quantification of two replicate experiments is shown below reporting the normalized signal relative to 37° for the 44° and 47° conditions (below) (Two-way ANOVA test was used to evaluate significance). (I) Representative western blots for the CETSA thermal stability assays as in panel H, for both GAPDH (control) and TBK1 in macrophages treated with DMSO or Momelotinib at 1x, 4x, or 10x the concentration used (left). The quantification of two replicate experiments is shown below reporting the normalized signal relative to 37° for the 44° and 47° conditions (below) (Dunnett's multiple comparison test was used to evaluate significance). Source data are available online for this figure.

(namely TFs) regulated by each CKI in a specific signaling context. At the same time, however, we acknowledge that the CKI space we explored was somehow limited and we expect that the inclusion of additional inhibitors targeting more kinases (such as those induced by macrophage activation) will result in further refinement of the analysis.

The data we obtained indicate clear divergences among drugs with similar intended targets (e.g., JAKi), with off-target effects that in general exceeded those that could be unambiguously assigned to the inhibition of the intended target, to the point that the spectrum of effects caused by each molecule was unique. Divergences from the expected results can likely arise from the incomplete knowledge of the targets of the kinases inhibited in vivo at the concentration used in the assay, but also from limitations of the in vitro profiling assays used to measure CKI specificity. A basic tenet of pharmacology is that in vitro binding affinities of small molecules to targets correlate with effects observed in cells, a notion that also applies to kinase inhibitors (Knight and Shokat, 2005). Indeed, kinobeads data were consistent with some otherwise unexpected effects of selected CKIs. For instance, the strong binding of the broad-spectrum CKI Midostaurin to TBK1, the main IRF3-activating kinase (Klaeger et al, 2017) accounted for its inhibitory effects on *Ifnb1* gene induction by LPS. Similarly, binding of the JAKi Momelotinib to TBK1 explained its similarity to Midostaurin in our analysis.

Other effects, however, were difficult to reconcile with in vitro binding data. For instance, the FDA approved JAKi Filgotinib, which was among the strongest inhibitors of STAT1 phosphorylation and IFN-mediated gene expression in our system, was not found to bind to any JAK family member in kinobeads data. Along the same line, Tofacitinib, a JAKi preferentially targeting JAK3, was found to inhibit STAT1 phosphorylation and Interferon-regulated genes in our system, yet it was not reported to bind any JAK family member in kinobeads-based assays (Klaeger et al, 2017).

Therefore, while a systematic assessment of in vitro binding affinities of CKIs is a requirement to describe the intrinsic ability of each molecule to interact with different components of the kinome,

it does not suffice to predict the extent of cellular perturbations caused by individual inhibitors.

Clearly, our approach is not suitable for the initial screening of large libraries of compounds but instead it can be applied to CKIs that have already advanced in the preclinical pipeline in order to determine the scope of their effects in living cells, as well as the similarities and differences among related molecules in a compound series. For instance, this approach would be most suitable to determine the extent of on- vs. off-target effects of molecules with similar activity towards the same kinase(s), particularly in combination with genetic approaches consisting in the measurement of CKI effects upon ablation of the intended target.

We anticipate that by further expanding the epigenomic datasets generated with multiple CKIs in different cell types and signaling contexts, and by integrating them with preclinical and clinical data on efficacy and side effects of CKIs, it will be possible to contribute to predict clinical outcomes of CKIs. Ultimately, this could aid in selecting molecules for further advancement into clinical trials.

## Methods

No statistical methods were used to predetermine sample size. Experiments were not randomized, and investigators were not blinded to allocation during the experiments or outcome assessment.

### Cell culture and reagents

Bone marrow isolation from male mice (age 6–8 wk, C57/BL6 strain from Charles River) was performed in accordance with Italian law (D.lgs. 26/2014), which enforces Directive 2010/63/EU (Directive 2010/63/EU of the European Parliament and of the Council of September 22, 2010, on the protection of animals used for scientific purposes). All animal procedures were approved by the Italian Ministry of Health (346/2017-PR, notification 75DA4.N.SXD). Mice were housed under specific pathogen-free (SPF) conditions at the Cogentech Mouse Genetics facility in

Milan, Italy, in individually ventilated cages (Tecniplast, Buguggiate, VA, Italy) with a 12-h light/dark cycle; food and water were provided ad libitum. Bone marrow-derived macrophages (BMDMs) were generated as described (Comoglio et al, 2019). BMDMs were treated with 10 ng/ml lipopolysaccharide from E. coli serotype EH100(Ra) (Alexis cat. ALX-581-010-L002) or 10 ng/ml IL-4 (Biotechne cat. 404-ML), DMSO or kinase inhibitors as listed in Dataset EV1. THP-1 cells (from ATCC code RRID:CVCL_0006) were cultured in RPMI 1640 supplemented with 10% North American serum, 1% pen/strep (Sigma P4333). Cell lines were authenticated by the Tissue Culture Facility of the European Institute of Oncology using the GenePrint10 system (Promega) and routinely screened for mycoplasma contamination. THP-1 monocytes were differentiated into macrophages by 24 h incubation with 10 ng/ml phorbol 12-myristate 13-acetate (PMA, Merck, P1585) followed by 48 h incubation in THP-1 culture medium. THP-1 were treated with 100 ng/ml LPS or 1000 U/ml recombinant Human Interferon-beta 1a (Tebu Bio cat. CLCYT236-2) or dsDNA with kinase inhibitors as listed in Dataset EV1. Double-stranded DNA was prepared as previously described (Chamma et al, 2022) using the following 45 bp ssDNAs:

5′-TACAGATCTACTAGTGATCTATGACTGATCTGTACAT GATCTACA,

5′-TGTAGATCATGTACAGATCAGTCATAGATCACTAGTA GATCTGTA.

Briefly, ssDNAs were annealed and its integrity verified by gel electrophoresis. dsDNA was delivered to THP-1 cells by lipofection using 2 µg of dsDNA and 2 µl of Lipofectamine RNAiMAX Transfection Reagent (ThermoFisher cat. 13778075) according to manufacturer's instructions. Cells were harvested 6 h after transfection.

## ATAC-seq

$5 \times 10^4$ BMDMs were lysed in 50 µL of 10 mM Tris-HCl (pH 7.4), 10 mM $MgCl_2$, and 0.1% Igepal CA-630. Nuclei were pelleted by centrifugation at $500 \times g$ for 20 min at 4 °C. The nuclear pellet was resuspended in 25 µL of a 10 mM Tris-HCl (pH 8.4) and 5 mM $MgCl_2$ buffer containing 1 µL of in-house produced Tn5 transposase (Picelli et al, 2014a). The reaction was incubated for 1 h at 37 °C and then stopped by adding 5 µL of cleanup buffer (900 mM NaCl, 300 mM EDTA), 2 ml of 5% SDS, and 2 µL of 20 µg/µL Proteinase K (Merck cat. P2308) and incubating the reaction for 30 min at 40 °C. Tagmented DNA was purified using 2X AMPure XP beads (Beckman cat. A63881). Finally, 2 µL of each of 10 µM forward and reverse dual indexing primers and 1 µl of KAPA HiFi HotStart (Roche cat. 07958889001) were used in a 50 ml reaction to PCR-amplify the tagmented DNA library. Fragments shorter than 600 bp were isolated by size selection (using 0.65X Agencourt AMPure XP beads) and then purified with 1.8X Agencourt AMPure XP beads. ATAC-seq library size was assessed using the Tapestation D5000 High-Sensitivity ScreenTape (Agilent cat. 5067-5592). Libraries were sequenced on an Illumina NextSeq 500 (76 bp single-end reads).

## H3K27ac ChIP-seq

H3K27ac ChIP-seq was carried out as previously described (Esnault et al, 2014) with minor modifications in order to multiplex ~1200 samples into 96-well plate formats. Briefly, $5 \times 10^6$ BMDMs

were pre-treated for 1 h with either DMSO or individual kinase inhibitors (concentrations used for each inhibitor are reported in Dataset EV1) and subsequently stimulated with either LPS or IL-4 for 0.5, 1, 2, and 4 h. Cells were then fixed with 1% formaldehyde (Merck cat. 252549) for 10 min at room temperature and thereafter quenched with 125 mM Tris-HCl pH 7.6. Fixed cells were washed 3 times with PBS and harvested by scraping and collected into a conical 96-well plate. Cells were then pelleted at 4000 rpm (Allegra X-12R centrifuge Beckman coulter) for 5′ at 4 °C; nuclei were extracted in 10 mM Tris pH 8, 10 mM NaCl, 0.2% NP-40, 1 mM EDTA, supplemented with protease inhibitors; pelleted nuclei were then washed in 10 mM Tris pH 8, 10 mM NaCl, 1 mM EDTA, supplemented with protease inhibitors; nuclei were permeabilized with SDS (0.3% final concentration) and incubating the cells at 65 °C for 5 min. This step allowed us to reduce sonication cycles. The SDS was quenched with 2% Triton X-100 for 30 min at 37 °C and the nuclei were then pelleted and resuspended into 100 µl of resuspension buffer (50 mM Tris pH 8, 1% SDS, 10 mM EDTA) supplemented with protease inhibitors. Chromatin was then sheared 12 samples at a time using the Diagenode Bioruptor pico (8 cycles, each 30″ on and 30″ off). 5 µl of sonicated samples were reverse crosslinked and purified through QIAquick 96 PCR Purification Kit (Qiagen cat. 28181) in order to evaluate the effective shearing on a 1.5% agarose gel. Sonicated samples were diluted 1:10 in 50 mM HEPES pH 8, NaCl 150 mM, 1% Triton X-100, 0,1% Sodium deoxycholate, supplemented with protease inhibitors. Lysates were incubated overnight at 4 °C with protein G Dynabeads (Invitrogen cat. 10009D) previously coupled with 2 µg of anti H3K27ac (Abcam Cat# ab4729, RRID:AB_2118291) antibody on a thermomixer within conical 96-deep-well plates (Thermo Scientific cat. 95040452) sealed with the Nunc™ 96 Well Caps (Thermo Scientific cat. 276011). After immunoprecipitation, beads were recovered using a 96 well magnetic stand and washed 6 times with RIPA buffer (50 mM Hepes pH 8, NaCl 150 mM, 1% Triton-X, 0.1% Na-deoxycholate, 0.1% SDS, 1 mM EDTA, supplemented with protease inhibitor cocktail), 4 times with a modified RIPA buffer containing LiCl (10 mM Tris pH 8, 0.25 M LiCl, 1 mM EDTA, 0.5% NP-40, 0.5% Na deoxycholate) and once in TE (10 mM Tris pH 8.0, 1 mM EDTA, 50 mM NaCl). Immunoprecipitated chromatin was eluted and cross-links reverted overnight at 65 °C. DNA was purified with QIAquick 96 PCR Purification Kit (Qiagen cat. 28181) and then quantified with Quanti-Fluor (Promega cat. E2670). Quality of individual ChIP samples was evaluated by qPCR using primers for the amplification of 3 distinct genomic loci: a negative control region to measure overall noise (chr12:94451782-94451875, Forward oligo: 5′-TTTTCCAGGCA AAGCAGATT; Reverse oligo: 5′-ATGTATGGGCACAAGCA CAA), an LPS-activated CRE (chr17:17901284-17901475, Forward oligo: 5′-TGGATTTCACATCACTTCACACA; Reverse oligo: 5′-G GTCACATCTTGCCACCACT) and an IL-4-controlled CRE (chr1:40322238-40322431, Forward oligo: 5′-ATTTGCCGCTCT ACCCAACA; Reverse oligo: 5′-TTGCTCAGTCAGGCAACCTC). Signal-to-noise from the real-time qPCR and the shearing size was determined in order to remove failed samples. Finally, ~600 ChIP-seq samples were selected for sequencing corresponding to a single full-time course per CKI per condition.

DNA libraries were prepared for NextSeq or Illumina NovaSeq 6000 sequencing as previously described starting from 2.5 ng of material (Austenaa et al, 2021). The purified DNA libraries were

quantified with the Quantifluor reagent (Promega cat. E2670) and quality-controlled using Tapestation (Agilent) with the high-sensitivity assay D5000 (Agilent cat. 5067-5592). DNA libraries were diluted to a working concentration of 4 nM and sequenced on a NextSeq or Illumina NovaSeq 6000 platform (76-bp or 51-bp reads).

## Chromatin-associated RNAs

$15 \times 10^6$ BMDM were pre-treated with DMSO for 1 h and then stimulated with either LPS (10 ng/ml) or IL-4 (10 ng/ml). Samples were retrieved at 0.5, 1, 2, and 4 h post-stimulation. Chromatin-associated RNA was obtained as described (Comoglio et al, 2019). For each time point, cells were lysed for 5 min with ice-cold lysis buffer (10 ng Tris-HCl pH 7.5, 0.15% NP-40, 150 ng NaCl). The lysate was then layered on 2.5 vol of a chilled sucrose cushion (24% sucrose in lysis buffer without NP-40) and centrifuged at 13,000 rpm for 10 min at 4 °C. Nuclei were gently rinsed with ice-cold PBS/1 ng EDTA and then resuspended in 200 ml of pre-chilled glycerol buffer (20 ng Tris-HCl pH 7.9, 75 ng NaCl, 0.5 ng EDTA, 0.85 ng DTT, 0.125 ng PMSF, 50% glycerol). An equal volume of a cold nuclear lysis buffer was added (10 ng HEPES pH 7.6, 1 ng DTT, 7.5 ng $MgCl_2$, 0.2 mM EDTA, 0.3 M NaCl, 1 M urea, and 1% NP-40). The tube was gently vortexed and then centrifuged at 13,000 rpm for 2 min at 4 °C. The chromatin pellet was rinsed twice with ice-cold PBS/1 mM EDTA and then dissolved in TRIzol (Invitrogen). RNase inhibitors (New England Biolabs, cat. M0307L) were added in all the buffers. Chromatin-associated RNA was purified with the Zymo Quick-RNA kit (Zymo Research, cat. R1055), according to manufacturer's protocol. Purified RNA was used for library generation using the Smartseq2 protocol as described (Gualdrini et al, 2022) with the following modification: 5 ng of total RNA was reverse-transcribed using 1 µl of 2 mM N6 (random primers 5'-AAGCAGTGGTAT CAACGCAGAGTACNNNNNN) instead of standard oligo-dT, and an LNA-containing template-switching oligo (TSO 5'-AAG CAGTGGTATCAACGCAGAGTACATrGrG+G-3'). Libraries were sequenced on an Illumina NextSeq 500 platform (76-bp single-end reads).

## RNA-seq

$10^6$ cells were pre-treated with DMSO or the individual kinase inhibitors for 1 h and then stimulated with LPS (10 ng/ml) for 2 h. Total RNA was extracted from three independent biological replicates using the Zymo Quick-RNA kit (Zymo Research cat. R1055) and subjected to a round of on-column DNA digestion, as per manufacturer's protocol. PolyA-RNA-seq libraries preparation was carried out using the SMART-seq2 protocol (Picelli et al, 2014b) with minor modifications. Briefly, 5 ng of total RNA was reverse-transcribed with template switching using oligo(dT) primers and an LNA-containing template-switching oligo (TSO). The resulting cDNA was pre-amplified and purified with the AMPure XP beads (Beckman cat. A63881). 2 ng of cDNA was tagmented with in-house made Tn5 transposase (Picelli et al, 2014a). cDNA fragments generated after tagmentation were gap-repaired and enriched by PCR with 1 µL of each 10 µM forward and reverse dual indexing primers and 1 µl of KAPA HiFi HotStart (Roche cat. 07958889001). The final cDNA library was purified

sequenced on an Illumina NovaSeq 6000 platform (51 bp paired-end reads).

## Flow cytometry

$10^6$ BMDMs were pre-treated with DMSO or the individual kinase inhibitors for 1 h at the designed $EC_{50}$ or $4xEC_{50}$ and then stimulated with LPS (10 ng/ml) for 2 h. Cells were washed twice in ice-cold PBS, gently scraped on ice and collected in a 96-well plate. Every centrifugation step was performed at $700 \times g$ for 5 min at 4 °C. Cells were pelleted and fixed by resuspension in BD Cytofix™ Fixation Buffer (BD cat. 554655) and incubation at 37 °C for 10 min. Subsequently, cells were washed once in staining buffer (1% BSA in PBS) and permeabilized by resuspension in 90% methanol:PBS. After 10 min incubation on ice, cells were washed twice in staining buffer to remove all traces of methanol and then blocked with 5% bovine serum albumin (BSA) in PBS for 30 min on ice. $10^6$ cells were then stained by incubation on ice for 1 h in the dark with 5 µl of primary antibody (Alexa Fluor® 647 Mouse Anti-Stat1 pY701, Clone 4a, BD Biosciences Cat# 612597, RRI-D:AB_399880) diluted in 100 µl 1% BSA in PBS and then washed twice in staining buffer before resuspending them in PBS for flow cytometry acquisition. Cells were analyzed using a BD FACSCe-lesta™ Cell Analyzer with the High Throughput Sampler (HTS) Option. Analyses were conducted using CytoExploreR (https://dillonhammill.github.io/CytoExploreR/). Per cell signal in individual replicate experiments ($n = 3$) were min–max normalized relative to the average Alexa Fluor 647 area of the DMSO condition (signal in DMSO untreated being the minimum and the 2 h LPS the maximum). Kernel density estimate, which is a smoothed version of the histogram, shows the combined signals of all min–max scaled samples. Box-plots are used to display the variability across biological triplicate experiments of the min–max scaled mode of the Alexa Fluor 647 area.

## Cellular thermal shift assay (CETSA)

The CETSA protocol was performed as described (Martinez Molina et al, 2013). $5 \times 10^6$ BMDMs were treated for 1 h with DMSO or kinase inhibitors. After incubation, cells were washed twice with ice-cold PBS and harvested by scraping in PBS supplemented with 1 mM (PMSF), a protease inhibitor cocktail (Roche cOmplete™, EDTA-free Protease Inhibitor Cocktail 4693132001), and kinase inhibitors or DMSO. Samples were pelleted at $3000 \times g$ for 3' at room temperature. Pellets were resuspended into PBS, mixed, and divided into 10 aliquots. Each aliquot was incubated at ten different temperatures (37, 41, 44, 47, 50, 52, 54, 56, 59, 63 °C) for 3 min, incubated 3 min at room temperature, and then snap-frozen. Cells were lysate by adding NaCl (final concentration 150 mM) and by performing freeze-thawing cycles. After cells lysis, samples were centrifuged at $17,000 \times g$ for 1 h at 4 °C and 30 µg of the supernatants, containing the soluble fraction of the proteome at each temperature, were loaded on 10% polyacrylamide gels and analyzed by western blot. Western blot signals (analyzed with Image Lab Software, Bio-RAD) for both TBK1 and GAPDH were individually scaled relative to the 37 °C signal and thereafter TBK1 was normalized relative to the GAPDH signal. Individual graphs in figures display temperatures of up to 50 °C, at which point GAPDH remains detectable.

## Western blot

$5 \times 10^6$ BMDMs were treated for 1 h with DMSO or kinase inhibitors and then treated for 15 min, 30 min, 1 h or 2 h with 10 ng/ml LPS, washed once in 1X PBS and then lysed in S300 buffer (50 mM Tris-HCl pH 7.6, 0.2% Nonidet P-40 (NP-40), 300 nM NaCl, 10% glycerol) supplemented with 1 mM PMSF, protease inhibitor cocktail (Roche cOmplete™, EDTA-free Protease Inhibitor Cocktail 4693132001), and PhosSTOP tablet (Phosphate Inhibitor cocktail Tablets Roche 4906837001) and incubated on ice for 10'. The solution was cleared by centrifugation at 13,000 rpm in microfuge for 10'. Protein extracts were resolved on SDS–polyacrylamide gel, blotted onto PVDF membranes, and probed with one of the following antibodies: Phospho-Stat1 (Tyr701) (D4A7) diluted 1:1000 from stock, Cell Signaling Technology Cat# 7649, RRID:AB_10950970; Phospho-p44/42 MAPK (Erk1/2) (Thr202/Tyr204) (D13.14.4E) diluted 1:2000 from stock, Cell Signaling Technology Cat# 12638, RRID:AB_265056; TBK1/NAK diluted 1:1000 from stock, Cell Signaling Technology Cat# 3013, RRID:AB_2199749; Phospho-SAPK/JNK (Thr183/Tyr185) (81E11) diluted 1:1000 from stock, Cell Signaling Technology Cat# 4668, RRID:AB_823588; IKBα (C-21) diluted 1:1000 from stock, Santa Cruz Biotechnology Cat# sc-371 (also sc-371-G), RRID:AB_2235952; Vinculin (7F9) diluted 1:10,000 from stock, Santa Cruz Biotechnology Cat# sc-73614, RRID:AB_1131294; GAPDH diluted 1:10,000 from stock, Abcam Cat# ab8245, RRID:AB_2107448.

## RT-qPCR

For RT-qPCR experiments, 500 ng of total RNA was reverse-transcribed with ImProm-II reverse transcription system (Promega A3800) following the manufacturer's instructions. RT-qPCR was assembled with Fast SYBR Green master mix (Applied Biosystems 4385614) and run on a QuantStudio 6 Pro real-time PCR machine (Applied Biosystems). Analysis was done on the Thermo Fisher Cloud platform. Primers used: C1orf43_fw GGATGAAAGCTCTGGAT GCC; C1orf43_rv GCTTTGCGTACACCCTTGAA; Ifnb1_fw TCC TCCAAATTGCTCTCCTGT; Ifnb1_rv TTCAATTGCCACAGGAG CTTC; Rsad2_fw TGGGTGCTTACACCTGCTG; Rsad2_rv GAAGT GATAGTTGACGCTGGTT.

## Analysis of chromatin-associated RNA-seq data

Single-end reads were mapped after adapter trimming to the mouse genome assembly mm10 (Illumina's iGenomes reference annotation downloaded from UCSC, http://support.illumina.com/sequencing/sequencing_software/igenome.html) and the RefSeq transcript annotation (ncbiRefSeqCurated November 16, 2017) using topHat2 (TopHat v2.1.1) (Trapnell et al, 2012) with parameters --max-multihits 2 --b2-very-sensitive. Multi-mapping reads and those mapping to the ENCODE blacklist regions (https://github.com/Boyle-Lab/Blacklist) (Amemiya et al, 2019) were removed using standard samtools and bedtools operations (Bedtools v2.29.2). Per-gene read counts were retrieved using standard R/Bioconductor packages (e.g., GenomicRanges and GenomicAlignment together with the proper GFF RefSeq annotation GRCm38.p6). Sample normalization was achieved by selecting invariant genes across samples/conditions using a strategy previously described (Gualdrini et al, 2016), see https://github.com/fgualdr/GeneralNormalizer). Differentially regulated genes across

time for both the LPS and IL-4 time-course were selected using ImpulseDE2 in case-only mode (Fischer et al, 2018).

## ATAC-seq data analysis

ATAC-seq samples were analyzed as previously described (Gualdrini et al, 2022). Briefly, after removal of adapter contaminations using BBDuk (https://github.com/BioInfoTools), single-end reads were trimmed with Trimmomatic (version 0.39 flags: EADING:3 TRAILING:3 SLIDINGWINDOW:4:15 MINLEN:38) (Bolger et al, 2014). Trimmed reads were then mapped to the mouse mm10 genome (Illumina's iGenomes reference annotation downloaded from UCSC, http://support.illumina.com/sequencing/sequencing_software/igenome.html) by exploiting the mapping steps of ShortStack (Johnson et al, 2016) (https://github.com/MikeAxtell/ShortStack). Reads unmapped, failing quality, or mapping to mitochondrial chromosomes were then removed using SAMtools (version 1.9). Peak calling was performed using MACS2 (version 2.2.6; options --nomodel --extsize 146 --nolambda --keep-dup "all" --call-summits). High confident reference set of peaks was created by selecting peaks called in each sample/replicate with q-value $\leq 10^{-10}$ and being consistent between replicates (overlapping area of at least 50% between replicates).

## TF ChIP-seq data analysis

### ATLAS harmonization

TFs ChIP-seq data were retrieved from the ChIP-ATLAS (https://ChIP-atlas.org) (Zou et al, 2022) and harmonized to the identified CREs from this study. Briefly, for each antibody and study (see Dataset EV4), peaks overlapping with at least one of the 59,617 CREs identified in this study and with an associated qValue $\leq 10^{-10}$ (computed using MACS2 according to the ChIP ATLAS description) were retained. Coverage per CRE for each TF ChIP-seq was retrieved from the deposited normalized BigWigs using the function ScoreMatrixList from the R Bioconductor packages genomation. We started from a total of ~400 samples and following several quality controls test which included number of peaks overlapping our reference ATAC-seq peaks, signal-to-noise from IGV-tracks and metaplots, we selected ~200 high-quality samples from 43 independent datasets and 34 different TFs.

## H3K27ac ChIP-seq mapping and normalization

Following adapter removal, single-end reads were first trimmed down to 51 bp read length mapped to the mouse mm10 genome using ShortStack (Johnson et al, 2016) (https://github.com/MikeAxtell/ShortStack). Read counts per sample around ATAC peaks (±700 bp from the summit) were retrieved using the R/Bioconductor packages GenomicRanges and GenomicAlignment (Lawrence et al, 2013). Sample normalization was achieved by selecting invariant peaks across samples as previously described (Gualdrini et al, 2016) (https://github.com/fgualdr/GeneralNormalizer). Briefly, we modeled the frequency distribution of the differences in read counts across samples with a Skewed-normal distribution. Peaks laying within 1σ of the best-fitted mean difference were considered as invariant and used to normalize the samples.

### *Identification and clustering of differentially regulated H3K27ac sites in LPS and IL-4-time courses*

Differentially regulated H3K27ac sites across time, for both the LPS and IL-4 time-course, were selected considering the unperturbed samples (DMSO pre-treated conditions) using ImpulseDE2 (R/Bioconductor package version 3.7; R version 3.6.2, abs(Log$_2$FC) > 1 and adjpval < 0.01). Density-based clustering was conducted on the matrix of read counts (considering significant signal-induced CREs) which was first centered and scaled. The UMAP algorithm (from the uwot R library see https://github.com/jlmelville/uwot) coupled to Louvain community detection (igraph R library, see https://igraph.org) was applied in order to extract macro groups/clusters in the unperturbed (DMSO) conditions.

## Multiple-factor analysis (MFA) applied to H3K27ac time course following CKI treatment

The experimental strategy adopted in this study involved the production of repeated time course under different perturbations (i.e., the individual CKIs). Given the size and complexity of the screen we aimed at selecting high-quality samples representing the full-time course per perturbation applied without replicates. Several statistical models have already been proposed to account for differential expression in the context of time series without replicates also exploiting dimensionality reduction (Albrecht et al, 2017; Spies et al, 2019; Wu and Wu, 2013). Within this study we developed a new empirical strategy to identify differentially regulated sites on the basis of pre-selected CREs significantly changing in time (see "*Identification and clustering of differentially regulated H3K27ac sites in LPS and IL-4 time course*"). We exploited the multi-table dimensionality reduction strategy "*Multiple Factor Analysis*" MFA in the sense of Escofier-Pagès (Escofier and Pagès, 2008), to simultaneously describe CREs kinetic and extract CKI-specific effects. MFA is an extension of the principal component analysis (PCA) for cases in which multiple quantitative tables are to be analyzed simultaneously (Abdi et al, 2013; Escofier and Pagès, 2008). At its core, MFA weighs the individual tables by the inverse of the first squared singular value of their PCA. By doing so, the first eigenvalue of the PCA performed on the weighted tables will be equal to 1 and therefore all the tables will play an equal role in the final representation, regardless of failed/discarded samples or outliers. Following this step, the individual weighted tables are combined and an additional non-weighted PCA is performed, which will lead to a common representation of the observations called *compromise* or *consensus*. Values on each dimension of the high-dimensional compromise space are termed *factor scores* and are mathematically equivalent to the barycenter, or averaged position, of the observations as seen by each individual data table termed *partial factor scores*. *Partial factor scores* are therefore the projection of the individual disjoint table onto the compromise space. The relative distance of the *partial factor scores* for each observation (or element in the data table, in this case the individual cis-regulatory elements selected as differentially controlled across time) is directly related to the differences among data tables.

Considering the datasets generated within this study, we structured the ~600 H3K27ac ChIP-seq samples by the stimuli used (LPS or IL-4) into 59 data tables according to the perturbation applied (the 58 individual CKIs, in addition to the DMSO, each containing the H3K27ac ChIP-seq read counts at each CRE at the various time points recorded). These tables were at first jointly normalized and standardized (i.e., centered and scaled) by CREs and subsequently the MFA algorithm applied (R-CRAN library FactoMineR version 2.7 (Lê et al, 2008). The analysis was conducted by stimuli therefore keeping the LPS and IL-4 conditions separate.

## Selection of significant MFA components

To select relevant dimensions of the compromise space, we performed over 500 random permutation and bootstrap replicates of the empirical data. In each iteration MFA was applied to the same multi-table dataset but with values permuted within columns (Camargo, 2022). Based on the bootstrap resampling and permutation a probability to observe same or greater variance by random per dimension was computed. Only those dimensions, obtained from the original data, with less than 1% chances of seen equal or greater variance by random were retained and considered for subsequent analysis (see Fig. 1D).

## Perturbation likelihood score

As mentioned above, within the MFA compromise space the *partial factor scores* are the coordinates associated to the single CREs when seen by the individual CKI and the DMSO tables. Consistent with these notions, the distance from the DMSO table for each CKI table at each CRE will reflect how different is the signal-induced change in H3K27ac signal considering all H3K27ac time points. We therefore computed the Euclidean distances from the DMSO table using the partial factor scores, as directly related to the deviation from the control condition. The Natural logarithm of the computed Euclidean distances were standardized (i.e., centered and scaled using mean and standard deviation computed by fitting a normal distribution; R-CRAN library fitdistrplus) and used to compute the complementary cumulative distribution function (CCDF, i.e., 1-CDF). While the computed probabilities will reflect the likelihood to observe by random a change relative to the DMSO condition as extreme as the one observed, it won't reflect the type of perturbation (i.e., increase or decrease in H3K27ac signal). We therefore assessed the relationship between the $-\text{Log}_{10}(\text{CCDF})$ and the differences across time between the DMSO and the individual CKI computed as follows:

$$DeltaZ = \sum_{t=0}^{tp}(zscoreCKI_t) - \sum_{t=0}^{tp}(zscoreDMSO_t)$$

Where *tp* are the individual time point: 0, 0.5, 1, 2, and 4 h (see Fig. 1E). The two values were combined to produce a sign-perturbation likelihood as:

$$signCCDF = -\log_{10}(CCDF) * \frac{DeltaZ}{|DeltaZ|}$$

This score was used to evaluate CKI proximity across the selected CREs (see below).

## Clustering CKIs on the basis of the sign-CCDF scores

In order to relate individual CKIs, we performed Principal Component Analysis (PCA) on the basis of the above computed

score. Relevant components were selected by bootstrap resampling as described for the MFA compromise components. KNN was computed within the high-dimensional PCA space and used to either perform CKI hierarchical clustering (using hclust function from the R library stats and *ward.D2* method) or as a 2D igraph considering only the 3 nearest connection per CKI (see https://igraph.org).

## Frequency distribution of CKI events across time

Given the two-dimensional space defined by the first two component of the MFA compromise, which relate to the kinetic of signal-induced changes of H3K27ac at the individual CREs, we computed the frequency distribution across the theta angle defined by these two axes. Theta angles in radians, defined by each CRE in the MFA compromise, were computed with the arctangent R function atan2 and converted to 0-360 degrees.

## Standardized Jaccard index

Jaccard indexes were computed between CKI effects (divided into up and down-regulatory events). To normalize for the different sizes of each group, individual indexes were standardized by considering the mean and standard deviation of randomly permuted CREs (total iteration of 1000 resampling). This approach account for the effect size of each group and is commonly termed standardized effect size (SES) of the Jaccard index.

## Association analysis between CKI effects and TFs peaks

To evaluate the significant association between CKI effects and TF ChIP-seq peaks (retrieved from the ChIP-seq ATLAS, https://ChIP-atlas.org) (Zou et al, 2022), we performed iterative permutation testing, exploiting the permTest function of the RegionR Bioconductor library to estimate the expected mean number of overlapping CREs with selected TFs ChIP-seq peaks under analysis (see individual Figures) (Gel et al, 2016).

## XGBoost classification machine learning and SHAP feature importance

To assess whether CKI-specific signal-induced H3K27ac perturbations could be predicted as a function of H3K27ac kinetics and TF binding, we used XGBoost, a gradient boosted decision tree-based classification algorithm (Chen and Guestrin, 2016). The original features used included: H3K27ac ChIP-seq time course following LPS stimulation (in unperturbed condition); H3K27ac ChIP-seq time course following IFNβ stimulation (GSE56121); MACS2 called peaks harmonized with our ~60,000 CREs from a selection of ~200 TFs ChIP-seq performed in primary BMDM in either resting or stimulated conditions (including LPS, LipidA, KLA etc., see Dataset EV4); signal per CREs of the ~200 TF ChIP-seq performed in primary BMDM in either resting or stimulated conditions (including LPS, LipidA, KLA etc., see Dataset EV4). Given the potential redundancy in the selected features we performed dimensionality reduction to mitigate multicollinearity and therefore extract orthogonal features to be feed to the XGBoost algorithm (Chen and Guestrin, 2016). MFA was adopted to compute a lower dimensional compromise space dividing the

datasets by antibody and individual study. MFA dimensions holding at least 1% of the variance were used as features to assemble XGBoost multi-class classifier models. ChIP-seq samples to MFA dimensions correlations or v-test (normally distributed criterion for categorical variables) were used to describe each component (see Fig. 3D).

Given the degree of unbalancing of the target variable to be predicted, custom weight has been computed giving overall higher importance to either up or down effects as opposed to the unaffected CREs (which compose the vast majority of the observations). Within changing observations higher weight has been given to that class holding more observations.

Briefly, the custom score was computed as follows:

Given the total number of observations ($n_{\text{samples}}$), the number of unaffected ($n_0$), the number of downregulated ($n_d$) and upregulated ($n_u$) CREs, we computed the weight for unaffected ($w_0$) and affected ($w_{\text{eff}}$) CREs as:

$$w_0 = \frac{n_{\text{samples}}}{(2 * n_0)} \text{ and } w_{\text{eff}} = \frac{n_{\text{samples}}}{(2 * (n_{\text{samples}} - n_0))}$$

The weight for the affected observations was scaled relative to the number of up ($n_u$) and down ($n_d$) observations:

$$w_u = \frac{n_u}{(n_d + n_u)} * w_{\text{eff}}; \; w_d = \frac{n_d}{(n_d + n_u)} * w_{\text{eff}}$$

A final $w_0$, $w_u$, and $w_d$ were then scaled so that their sum added up to 1:

$$w_{us} = \frac{W_u}{(W_u + W_d + W_0)}; \; w_{ds} = \frac{W_d}{(W_u + W_d + W_0)}; \; w_{0s} = \frac{W_0}{(W_u + W_d + W_0)}$$

In this way, we would assemble models in favor of observable effects giving higher importance to the main effect observed. When the frequency of up and down regulatory effects is equal to the weight it is also equally distributed. Per observation weights were provided during the XGBoost fitting step as "sample_weights" in the "fit_params".

The dataset comprising the ~16,500 LPS-controlled CREs was split into training and testing set with a ratio of 70% training and 30% testing, and stratified by the target variable (this was done to ensure equal representation of the sparse data to be predicted). XGboost hyperparameter tuning was achieved by running grid-search cross-validation (GridSearchCV), with a 3 by 3 repeated stratified Kfold CV and with the following settings: subsample = [0.8]; objective = ['multi:softprob']; colsample_bytree = [0.8]; gamma = [0.01,0.1,0.5,1,2]; reg_alpha = [0.1,0.2,0.4,0.8]; reg_lambda = [0.1,0.5,1,1.5,2]; max_depth = [3,6,9]; max_leaves = [0]; max_bin = [30, 60, 90, 150, 200, 250, 300]; learning_rate = [0.05, 0.07, 0.1]; n_estimators = [50,150,300,600]; booster = ['gbtree']; tree_method = ['exact']; n_jobs = [4]; sampling_method = ['gradient_based']; base_score = [0.5]; min_child_weight = [4]; default_direction = ['learn']; grow_policy =['depthwise','lossguide']. In total 9 folds cross-validations were run against 84,000 hyper parameters candidate combinations, totaling 756,000 fittings for each model. Parameter selections were evaluated on the basis of the average of recall obtained on each class (i.e., balanced accuracy). The selected models were then evaluated on the test set reporting both ROC-AUC and PR-AUC. Shapley additive explanations (SHAP) (Lundberg and Lee, 2017) is a game theoretic approach for interpreting predictions of complex

models providing interpretable predictions for a test sample. To extract relative feature importance, we applied the SHAP-TreeExplainer algorithm, a SHAP based explanation method for trees enabling computation of local explanations (Lundberg et al, 2020). The mean feature importance was calculated as the average of the SHAP values magnitudes across each dataset and scaled to range between 0 and 1 in order to plot relative importance and compare the 58 independent models. To include the type of impact each feature had on each model, we computed the correlation between each feature and the corresponding SHAP value. The signs of the correlation coefficients (positive or negative) were multiplied by the computed averaged scaled SHAP values which we termed "*directional SHAP importance*" (see Fig. 3E).

Python workflow to run both XGboost classifier and SHAP feature importance is available at: https://github.com/fgualdr/MultiClassXGBOOST.

### RNA-seq data analysis

Single-end reads were mapped after adapter trimming to the mouse genome assembly mm10 (Illumina's iGenomes reference annotation downloaded from UCSC, http://support.illumina.com/sequencing/sequencing_software/igenome.html) and the RefSeq transcript annotation (ncbiRefSeqCurated November 16, 2017) using topHat2 (TopHat v2.1.1) (Trapnell et al, 2012) with parameters --max-multihits 1 --b2-very-sensitive. Reads mapping to the ENCODE blacklist regions (https://github.com/Boyle-Lab/Blacklist) (Amemiya et al, 2019) were removed using standard Bedtools operations (Bedtools v2.29.2). Per-gene read counts were retrieved using standard R/Bioconductor packages (e.g., GenomicRanges and GenomicAlignment together with the proper GFF RefSeq annotation ncbiRefSeqCurated, November 16, 2017). Sample normalization was achieved by selecting invariant genes across samples/conditions (Gualdrini et al, 2016) similarly to the strategy applied to ChIP-seq and ATAC-seq. Differentially regulated genes were selected using DESeq2 (R/Bioconductor package version 1.26.0; R version 3.6.2) after turning off the default normalization that DESeq2 applies. Squire (Yang et al, 2019) was implemented to evaluate Transposable elements expression (Family-wide) from the RNA-seq data. Gene set enrichment analysis was conducted against the MSigDB (msigdb_v7.5.1) computing the hypergeometric enrichment and the Standardized Jaccard index.

## Data availability

The datasets and computer code produced in this study are available in the following databases: Chip-seq data: Gene Expression Omnibus: GSE219239. ATAC-seq data: Gene Expression Omnibus: GSE218966. RNA-seq data: Gene Expression Omnibus: GSE219229. Chromatin-associated RNA-seq: Gene Expression Omnibus: GSE218968. Modeling computer scripts: GitHub (https://github.com/fgualdr/CKISCREEN_analysis_2022 and https://github.com/fgualdr/MultiClassXGBOOST). ChIP-seq: BED and BigWIG files. The links and accession numbers for the 189 datasets used are listed in Dataset EV4. All these datasets were downloaded from the ChIP Atlas (https://chip-atlas.dbcls.jp/data/mm10/eachData/).

The source data of this paper are collected in the following database record: biostudies:S-SCDT-10_1038-S44320-024-00040-x.

## Peer review information

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

## Acknowledgements

This project was funded by the European Commission (Advanced ERC grant #692789 to GN and the Marie Sklodowska-Curie Actions fellowship "MetChromTx" ID 789792 to FG). This work was also partially supported by the Italian Ministry of Health with the "Ricerca Corrente" and "5×1000" funds to the IEO IRCCS. FP is a PhD student in the European School of Molecular Medicine (SEMM) and is supported by a fellowship from the Associazione Italiana Ricerca sul Cancro (AIRC). We thank the members of the IEO Genomics Unit (Luca Rotta, Thelma Capra, Giulia Sabbatinelli) for support in datasets generation; Arnaud Ceol of the IEO Computing, Data and Digital Research Platforms Unit for help with data storage and analysis; Silvia Monticelli (IRB, Bellinzona, CH) for critical comments on the manuscript.

## Author contributions

**Francesco Gualdrini:** Conceptualization; Data curation; Software; Formal analysis; Supervision; Funding acquisition; Investigation; Visualization; Methodology; Writing—original draft; Writing—review and editing. **Stefano Rizzieri:** Data curation; Investigation; Methodology. **Sara Polletti:** Supervision; Investigation; Methodology; Writing—original draft; Writing—review and editing. **Francesco Pileri:** Investigation; Methodology; Writing—review and editing. **Yinxiu Zhan:** Investigation; Methodology. **Alessandro Cuomo:** Supervision; Investigation; Methodology. **Gioacchino Natoli:** Conceptualization; Supervision; Funding acquisition; Investigation; Writing—original draft; Writing—review and editing.

Source data underlying figure panels in this paper may have individual authorship assigned. Where available, figure panel/source data authorship is listed in the following database record: biostudies:S-SCDT-10_1038-S44320-024-00040-x.

## Disclosure and competing interests statement

The authors declare no competing interests.

