## [Peer Review File · Molecular Systems Biology]

An integrative epigenome-based strategy for unbiased profiling of clinical kinase inhibitors

Gioacchino Natoli, Francesco Gualdrini, Stefano Rizzieri, Sara Polletti, Francesco Pileri, Yinxiu Zhan, and Alessandro Cuomo

Corresponding author(s): Gioacchino Natoli (gioacchino.natoli@ieo.it) , Francesco Gualdrini (francesco.gualdrini@ieo.it)

Review Timeline:

Submission Date:	28th Oct 23
Editorial Decision:	21st Dec 23
Revision Received:	26th Feb 24
Editorial Decision:	8th Apr 24
Revision Received:	12th Apr 24
Editorial Decision:	15th Apr 24
Revision Received:	16th Apr 24
Accepted:	18th Apr 24

Editor: Poonam Bheda

Transaction Report:

21st Dec 2023

Manuscript Number: MSB-2023-12088

Title: An integrative epigenome-based strategy for unbiased functional profiling of clinical kinase inhibitors

Dear Dr. Natoli,

Thank you again for submitting your work to Molecular Systems Biology. We have now heard back from the three reviewers who agreed to evaluate your study. As you will see from the reports below, the referees acknowledge the interest of the study and are overall supportive of your work; however they also comment on multiple aspects of the manuscript that should be strengthened in a revision.

Without repeating all the comments listed below, some of the more fundamental issues raised are the following:

- experimental setup needs to be better described, as do the bioinformatics, and the conclusions should be more logically explained
- the limitations and any confounding factors of the method and analyses should be discussed, as Reviewer 1 pointed out on the polypharmacology of the tested CKIs
- the limitation on the applicability of the conclusions to other species and cell types, as pointed out by Reviewer 2. While discussion of the limitations may be sufficient, we would suggest at least minimal validation in some human cells, given an aim of the study is for the relevance of CKIs in clinical settings.

All other issues raised would need to be satisfactorily addressed. Please let me know in case you would like to discuss in further detail any of the comments, I would be happy to schedule a call.

We require:

1) A .docx formatted version of the manuscript text (including legends for main figures, EV figures and tables). Please make sure that the changes are highlighted to be clearly visible. Alternatively you may choose to submit your manuscript as a LaTeX file.

4) A .docx formatted letter INCLUDING the reviewers' reports and your detailed point-by-point responses to their comments. As part of the EMBO Press transparent editorial process, the point-by-point response is part of the Review Process File (RPF), which will be published alongside your paper.

5) A complete author checklist, which you can download from our author guidelines (<https://www.embopress.org/page/journal/17574684/authorguide#submissionofrevisions>). Please insert information in the checklist that is also reflected in the manuscript. The completed author checklist will also be part of the RPF.

6) Please note that all corresponding authors are required to supply an ORCID ID for their name upon submission of a revised manuscript.

7) It is mandatory to include a 'Data Availability' section after the Materials and Methods. Before submitting your revision, primary datasets produced in this study need to be deposited in an appropriate public database, and the accession numbers and database listed under 'Data Availability'. Please remember to provide a reviewer password if the datasets are not yet public (see <https://www.embopress.org/page/journal/17574684/authorguide#dataavailability>).

This study includes no data deposited in external repositories.

8) For data quantification: please specify the name of the statistical test used to generate error bars and P values, the number (n) of independent experiments (specify technical or biological replicates) underlying each data point and the test used to calculate p-values in each figure legend. The figure legends should contain a basic description of n, P and the test applied.

Graphs must include a description of the bars and the error bars (s.d., s.e.m.). Please provide exact p values.

<https://www.embopress.org/page/journal/17574684/authorguide#expandedview>

11) For more information: There is space at the end of each article to list relevant web links for further consultation by our readers. Could you identify some relevant ones and provide such information as well? Some examples are patient associations, relevant databases, OMIM/proteins/genes links, author's websites, etc...

12) Author contributions: CRedit has replaced the traditional author contributions section because it offers a systematic machine readable author contributions format that allows for more effective research assessment. Please remove the Authors Contributions from the manuscript and use the free text boxes beneath each contributing author's name in our system to add specific details on the author's contribution. More information is available in our guide to authors.

13) Disclosure statement and competing interests: We updated our journal's competing interests policy in January 2022 and request authors to consider both actual and perceived competing interests. Please review the policy <https://www.embopress.org/competing-interests> and update your competing interests if necessary.

14) Every published paper now includes a 'Synopsis' to further enhance discoverability. Synopses are displayed on the journal webpage and are freely accessible to all readers. They include a short stand first (maximum of 300 characters, including space) as well as 2-5 one-sentences bullet points that summarizes the paper. Please write the bullet points to summarize the key NEW findings. They should be designed to be complementary to the abstract - i.e. not repeat the same text. We encourage inclusion of key acronyms and quantitative information (maximum of 30 words / bullet point). Please use the passive voice. Please attach these in a separate file or send them by email, we will incorporate them accordingly.

Please also suggest a striking image or visual abstract to illustrate your article as a PNG file 550 px wide x 300-600 px high. Share synopsis text and image, as well as eTOC:

Please note that these would be the final versions and changes during proofing are usually not allowed

15) As part of the EMBO Publications transparent editorial process initiative (see our Editorial at <http://embomolmed.embopress.org/content/2/9/329>), Molecular Systems Biology Medicine will publish online a Review Process File (RPF) to accompany accepted manuscripts.

In the event of acceptance, this file will be published in conjunction with your paper and will include the anonymous referee reports, your point-by-point response and all pertinent correspondence relating to the manuscript. Let us know whether you agree with the publication of the RPF and as here, if you want to remove or not any figures from it prior to publication.

Molecular Systems Biology has a "scooping protection" policy, whereby similar findings that are published by others during review or revision are not a criterion for rejection. Should you decide to submit a revised version, I do ask that you get in touch after three months if you have not completed it, to update us on the status.

I look forward to receiving your revised manuscript.

Yours sincerely,

Poonam Bheda

Poonam Bheda, PhD
Scientific Editor
Molecular Systems Biology

Use the link below to submit your revision:

Reviewer #1:

Review of MSB-2023-12088

In this manuscript, Gualdrini et al developed a ChIP assay that measures the response of the epigenome (using acetylated H3K27 as a proxy) of activated macrophages to the treatment by kinase inhibitors. This is quite an interesting assay as it complements other omics-type drug characterization assays that are focused on e. g. target binding, pathway engagement, and cellular reprogramming. That said, the manuscript needs a lot of work before it may be acceptable for publication in MSB. First, the introduction needs better coverage of the relevant literature. Second, the authors must more clearly describe their experimental setup so that readers know which experiments were done. Third, the bioinformatics-heavy section are largely unintelligible and require a major re-write. Fourth, it has to come out more clearly how the authors came to the concluding remarks in each section. I could not follow this in most instances. Fifth, and perhaps most important, the authors must discuss to what extent their findings can and should be interpreted for the intended purpose. In other words, to what extent are the results in the paper the consequence of polypharmacology. I sometimes had the impression that the authors tried too hard to ascribe certain effects to a particular drug mechanism when the more obvious explanation is found in the fact that almost none of the drugs used is pharmacologically clean.

Specific comments:

Introduction: the authors focus the introduction on methods that are useful for deconvoluting the targets of kinase inhibitors. This is of course one important relevant aspect for the work presented in this study but falls short of covering further important steps before any signal arrives at the level of transcription factors. The authors should mention how the signal transduction can be measured inside cells in response to KIs. For this, there is a rich body of (also very recent) literature on e.g. phosphoproteomic response of cells to KIs that illuminate their cellular mechanism of action. I strongly suggest that this aspect is covered in the introduction and discussion.

Introduction: the authors should avoid statements such as "unbiased functional informatoin" given that their approach is very focussed on epigenetic effects. And here, it is overclaiming to say that "...systems level analysis of histone acetylation changes... allow determining which signaling pathways are affected..." such changes in histone acetylation will affect so many parts of chromatin that deconvoluting this to individual genes etc is not trivial. It would be better to exchanged "unbiased" to something like "focussed or specific" to bring out that the analysis of histone acetylation is only one readout and actually a summarized proxy for changes in chromatin/transcription.

Results:

General remark: there is rather much introductory text in some sections.

Section on "Histone acetylation as unbiased readout of CKIs' effects.": it would be useful to state in this first section which CKIs are used. As for the concentration: why did the authors use the CATDS metric? This was derived from data collected in lysates not in cells. Hence, this metric is not necessarily useful for performing cellular experiments. It would be much stronger to measure the CKIs in a dose-dependent fashion which would be directly relevant for a cellular assay. The final choice of a CATDS of 0.1 is unclear. This would actually include pretty unselective molecules. And why use a different criterion for TLR4 or IL-4R signaling? Why just one antibody for ChIP-seq? Please be more specific here and elsewhere: "by generating several hundred H3K27ac ChIP-seq data sets" How many and how do these come about? MFA analysis: this text is rather inaccessible to a common reader and I struggled in many places to follow the descriptions/arguments. Please simplify, use clearer and more consistent language and focus the text on what the MFA analysis shows with regard to the purpose of the experiment. I am surprised by the statement that the shift in the barycenter is a quantitative measure. It is not in PCA, so how come it is in MFA? Does the shift in barycenter correlate with the number of targets of the CKI used? If yes, how can one attribute the effects to a particular kinase? What is an MFA dimension? Why was it necessary to compute a perturbation likelihood when your assay is supposed to measure perturbation directly? It is unclear to me what the purpose of the 2D network-graph is. And I do not see a justification for the conclusion that "...effects of individual CKIs are influenced by the underlying network of active signaling pathways...". It is much more likely that the lack of such correlation is rooted in the polypharmacology of the drug (i.e. more than one potent target). This is a fundamental issue in this analysis at least in the way the authors are trying to portray it. Up to this

point it is not at all clear how the generated data can be meaningfully interpreted.

Section on "Proximity among CKIs as a function of kinetics and genomic overlap of H3K27ac changes." I could repeat many of the comments above. Please find a way to state the purpose of the analysis and formulate the results in a way that can be interpreted. It starts with the title of the section: what is "proximity among CKIs"? From my read of the text, it is entirely unclear how the authors arrive at this conclusion: "Altogether, these observations suggest that CKIs proximity based on H3K27ac changes, reflects the global spectrum of actions of the individual inhibitors." At no point in this section do the authors state what the action of these inhibitors actually is. What are the transcriptional programs they do or do not affect? Do these bear any relationship with known cellular consequences of these drugs in the chosen biological system?

Section on "Deconvolution of CKI effects by identification of CKI-regulated transcription factors.": This reviewer failed to gather what the authors are trying to achieve in this section and the results of the ROC deeply concern me. Also, Midostaurin is among the least selective CKIs we know today. So how can one envisage using an epigenetic profile to predict what the drug is going to do? If this is not the intent of the analysis, this reviewer must be missing an important point.

Section on "Distinctive and unique properties of individual JAK inhibitors (JAKi)": This section of the manuscript was easier to follow but the results are trivial and the interpretation obvious. The fact that these JAK inhibitors have other targets, sometimes potent, should place them in all kinds of areas of the CKI effect space. What is more relevant though is if the biological system has active signaling components that utilize these targets. This section highlights an issue in the overall approach in that the authors used in-lysate kinobead data and poor selectivity compounds to try and design a cellular perturbation experiment. One cannot assume that the potency of kinobead binding translates into a similar potency in cells. This will have to be assessed on a case by case basis and may well explain why some of the JAK inhibitors did not do anything to STAT phosphorylation in the flow assay at the presumed EC50 from kinobeads. Just a note, the kinobead data reports apparent dissociation constants, not EC50 values.

Section on "Unexpected effects of CKIs on the IFN β -regulated gene expression program." The results of the GO analysis are obvious insofar as macrophages are cells of the immune system and changes in their epigenomes/transcriptional profiles will always produce strong GO terms related to the basic underlying biology. The same would be observed in eg brain or liver cells. This is a well known limitation of current GOs and cannot be taken as evidence the way suggested by the authors. There is a misconception in this paragraph regarding the CETSA assay. Although widely portrayed as such, drug binding does not necessarily increase the thermal stability of its target. The number of cases that do the opposite is just as high and many drugs do not show any thermal shift. CETSA shifts just record differences in the energy landscape of/around the protein that may be a direct or indirect consequence of drug binding. But yes, the results of the authors imply that their assay can be more informative than a CETSA.

Reviewer #2:

Gualdrini et al present an innovative study with massive amounts of data, profiling the epigenomic effects of 58 clinically approved kinase inhibitors. Each inhibitor is tested in affecting the LPS- and/or IL4-response of macrophages, characterized by a short timecourse of H3K27Ac, RNA-seq data at the 2 hr timepoint, and finally ChIP-seq studies of dozens of stimulus-induced transcription factors. This is a uniquely comprehensive dataset that has the potential to yield novel insights.

The dataset also poses some data analysis challenges. The manuscript details the rationale of the data analysis workflow and the description seems comprehensible to a fairly general/informed audience and makes sense. At the same time there is no doubt that alternative analysis workflows are possible, and these may lead to other insights.

The present analysis is largely restricted to learning about the specificity or breadth of the pharmacological profile of each CKI, and comparing it to in vitro data. This makes sense and some insights of general interest are learned. (The dataset may also be of interest for other questions, for example from the perspective of predicting gene expression and gene regulatory mechanisms, with the CKIs being used as perturbations to develop a model.)

As a result there is also a wealth of granular information about the specificity or overlap of particular CKIs that may be of interest to small numbers of researchers. Some of these are worked through to gain mechanistic detail.

Overall this is an impressive and novel study that deserves to be published. While the insights hold some interest, the datasets have value in their own right as they allow many further studies.

The manuscript is well written. Here some minor points:

58 CKI were chosen: why are they described as non-overlapping: when some target the same kinase, and 238 pairs shared more than one target? This is quite remarkable and it would be nice to see this enumerated in the main text/figure.

p.4 homogeneous criteria => consistent criteria

IL4 - not sure that referring to it as a homeostatic stimulus makes sense. It is a microenvironmental cytokine that also elicits macrophage responses.

p.8 Defined as the size of the intersection of the CREs affected by two CREs => CKIs

Reviewer #3:

This manuscript (An integrative epigenome-based strategy for unbiased functional profiling of clinical kinase inhibitors) from the Natoli lab presents an integrative analytical approach to identify similarities and differences among clinical kinase inhibitors (CKIs) with varying specificities, based on chromatin modifications (H3K27ac). Specifically, the authors use the well known model of inflammatory responses in bone marrow derived macrophages stimulated either with LPS or IL4 for different time frames.

This is an interesting, novel systems biology approach to compare the effects of the different compounds on a global scale, at the genomic/transcriptional level. In general, the data is well presented and of interest to a broad audience. I would therefore recommend publication in Mol Syst Biol, once the following comments have been addressed:

- The authors observed that H3K27ac marks had a substantial degree of overlap in specificities among CKIs, but no such overlaps were seen in the RNAseq computed model (Figure 2). (Could this be due to differences in treatment/time points?) Moreover, several inhibitors unexpectedly maintained the activation of immune response upon LPS stimulation (Figure 5). Could the authors please discuss and elaborate on their interpretation.

- I may have missed or overlooked this part, but did the authors check and present CKIs that had no effects, i.e. that did not have effects on H3K27ac levels under the conditions tested?

- Also, in general, H3K27ac patterns and TF activities are lineage and cell type dependent. The findings presented here are an excellent example of innate immune responses in murine macrophages, but of course, validation in human cells or in other cell types would be required to generalize the conclusions drawn here.

- The selection criteria for the kinase inhibitors included their targets being components of the TLR4/IL-4R signaling pathways, as well as the kinases being transcriptionally activated in response to LPS or IL-4. Obviously, more kinases or rather CKIs could be added to the list, given that many of them may be regulated at the post-translational level (protein expression, PTMs). This point may be emphasized in the discussion section, if it is not feasible to add more data.

Mior comments:

- page 5, paragraph 1; the last line appears to be an incomplete sentence (after "and the PI3 kinase").

- page 5, paragraph 3, line 2-3; it's mentioned "and about half of them could be assigned to a total of 208 CKIs with a wide range of selectivity scores". However, Figure 1 illustrates 177 CKIs in total. Please correct this inconsistency.

- Page 5, paragraph 3, line 4-5; the text mentioned that "with a CATDS > 0.5, eventually resulting in a list of 24 inhibitors". However, Figure 1 shows 23 inhibitors. Please correct.

MSB-2023-12088 – Gualdrini et al. *An integrative epigenome-based strategy for unbiased functional profiling of clinical kinase inhibitors*

We thank the reviewers for the extensive and constructive feedbacks to our study. To address all the reviewers' concerns we have extensively modified the description of the methodological approach and the main findings of our study to make them more comprehensible and readable. We have also included additional analyses and experimental validation in human cells, as detailed below. We believe that the paper is much improved as a result of these revisions and we are grateful to the reviewers for their valuable contributions.

Reviewer #1:

"In this manuscript, Gualdrini et al developed a ChIP assay that measures the response of the epigenome (using acetylated H₃K₂₇ as a proxy) of activated macrophages to the treatment by kinase inhibitors. This is quite an interesting assay as it complements other omics-type drug characterization assays that are focused on e. g. target binding, pathway engagement, and cellular reprogramming.

That said, the manuscript needs a lot of work before it may be acceptable for publication in MSB.

First, the introduction needs better coverage of the relevant literature.

Second, the authors must more clearly describe their experimental setup so that readers know which experiments were done.

Third, the bioinformatics-heavy section are largely unintelligible and require a major re-write.

Fourth, it has to come out more clearly how the authors came to the concluding remarks in each section. I could not follow this in most instances.

Fifth, and perhaps most important, the authors must discuss to what extent their findings can and should be interpreted for the intended purpose. In other words, to what extent are the results in the paper the consequence of polypharmacology. I sometimes had the impression that the authors tried too hard to ascribe certain effects to a particular drug mechanism when the more obvious explanation is found in the fact that almost none of the drugs used is pharmacologically clean."

We would like to thank this reviewer for appreciating the relevance of our study and for providing extensive and constructive feedbacks for improvement, particularly with respect to the writing style and the description of the methodological approach and the main findings. All the issues raised by this reviewer have been addressed in the revision, as detailed in the response to the specific comments below.

Specific comments:

"Introduction: the authors focus the introduction on methods that are useful for deconvoluting the targets of kinase inhibitors. This is of course one important relevant aspect for the work presented in this study but falls short of covering further important steps before any signal arrives at the level of transcription factors. The authors should mention how the signal transduction can be measured inside cells in response to KIs. For this, there is a rich body of (also very recent) literature on e.g. phosphoproteomic response of cells to KIs that illuminate their cellular mechanism of action. I strongly suggest that this aspect is covered in the introduction and discussion."

We appreciate the insightful feedback from the reviewer and acknowledge the importance of citing in the introduction highly relevant approaches and in particular phosphoproteomics. We have now incorporated relevant recent papers in our introduction (see rows 102-110). We have also clarified both in the introduction and the discussion sections how our approach is meant to complement and not replace biochemical approaches to CKI profiling.

"Introduction: the authors should avoid statements such as "unbiased functional informatoin" given that their approach is very focussed on epigenetic effects. And here, it is overclaiming to say that "...systems level analysis of histone acetylation changes... allow determining which signaling pathways are affected..." such changes in histone acetylation will affect so many parts of chromatin that deconvoluting this to individual genes etc is not

trivial. It would be better to exchanged "unbiased" to something like "focussed or specific" to bring out that the analysis of histone acetylation is only one readout and and actually a summarized proxy for changes in chromatin/transcription."

We thank the reviewer for these suggestions. We have made the necessary adjustments in the text (e.g., row 113). Our use of "unbiased" was meant in opposition to analyses focusing on narrow readouts, which come with the risk of providing very partial views on CKIs' effects. We have now replaced "unbiased readout" with "specific and interpretable readout". We do recognize the challenges of unraveling the multitude of effects induced by individual CKIs on signaling pathways through genome-wide time-resolved H3K27ac changes. However, we wish to stress the impartial nature of our assay and its potential to identify, because of its high granularity, similarities and differences among CKIs that may not be clear using other profiling approaches.

"Results:

General remark: there is rather much introductory text in some sections."

We do agree and we have amended the text accordingly.

"# Section on "Histone acetylation as unbiased readout of CKIs' effects.": it would be useful to state in this first section which CKIs are used. As for the concentration: why did the authors use the CATDS metric? This was derived from data collected in lysates not in cells. Hence, this metric is not necessarily useful for performing cellular experiments. It would be much stronger to measure the CKIs in a dose-dependent fashion which would be directly relevant for a cellular assay."

We thank this reviewer for giving us the opportunity to provide additional clarifications on these important issues. The comprehensive list of the 58 individual cyclin-dependent kinase inhibitors (CKIs), along with very extensive descriptions and annotations, has been included in the first supplementary table (**Dataset EV1**) and **Appendix Figure S1**. In the interest of conciseness and readability, we refrained from indicating the full names of all the 58 compounds within the manuscript body.

Regarding the concern raised about the use of the CATDS metric and its relevance to cellular experiments, we do understand this issue but we would like to emphasize that the choice of using the CATDS metric was a very extensively pondered decision.

First, the CATDS metric, albeit obtained in lysates, provided a uniform metric for all the 243 CKIs that could be measured with this approach (Klaeger and colleagues, 2017). In fact, no comparable data set is available that could have been used instead.

Second, as the CKIs used were all in different phases of clinical development, they were all optimized for their high ability to cross cell membranes, which is expected to make possible discrepancies between in vitro and in vivo EC₅₀ rather negligible.

Third, each concentration employed, reflecting the apparent dissociation constant in cell lysates, was cross-referenced with available (yet non-homogeneous) data on concentrations used to treat intact cells in various publications. This information is directly available in the **Dataset EV1** together with the relevant PubMed IDs used as references and explicitly commented in the results. For the majority of the cases, the concentration used in our assay either exceeded or was within the range of concentrations reported to cause detectable effects in intact cells. We have made the necessary adjustments in the text (see row 214-219).

In addition, no correlation between the total number of effects in H3K27ac changes for either LPS or IL4 signaling with the concentration used was observed (see **Figure 1** below). These observations align with the reviewer's comments acknowledging that we recognize the presence of multiple strong targets for each compound. We do not at all dismiss the possibility that the known polypharmacology of these compounds complicates the interpretation of our data. However, it is crucial to emphasize that in the context of such multi-target spectra, the underlying signaling network plays a vital role in determining the phenotypic consequences observed. The combination of polypharmacology and the specific signaling context collectively influences the interpretation of our results.

Clearly, the use of dose-response analyses for each CKI would have been ideal. However, it is essential to underscore the feasibility issues associated with this scenario when considering the intrinsic low throughput of ChIP-seq assays. The time-dependent activation of various transcription factors (TFs) in response to lipopolysaccharide (LPS) and interleukin-4 (IL₄) stimulation necessitates a nuanced, time-resolved kinetic approach. Furthermore, conducting a dose-response experiment with multiple concentrations (typically ranging from 7 to 10 doses, as per recent literature) becomes logistically challenging to the point of being not feasible because of the massive number of samples to be generated and analyzed.

Figure 1. Comparison of the number of detected effects in H₃K₂₇ac changes driven by each CKI (x-axis) with the reported concentration in the literature (y-axis). The color of each dot represents the discrepancy (fold 'used' over 'reported' in literature) between the dose used in our assay and the one from the literature. The analysis was performed for both LPS (left) and IL₄ (right) conditions.

“The final choice of a CATDS of 0.1 is unclear. This would actually include pretty unselective molecules. And why use a different criterion for TLR₄ or IL-4R signaling?”

We appreciate the opportunity to provide the opportune clarifications on this important matter. The selectivity criteria for CKIs within both the IL-4 and LPS signaling contexts were, in fact, identical, with a cutoff of 0.1 applied to both. While the ideal scenario would have involved the inclusion of CKIs exhibiting relatively high selectivity, the need to target a broad spectrum of signaling components forced us to lower the stringency of our selection criteria. Indeed, even applying a lenient cutoff of 0.1, we identified only 36 CKIs, which targeted a partially overlapping and redundant set of targets encompassing 24 components of the two signaling pathways. In addition to these 36 inhibitors, we introduced additional CKIs predicted to impact kinases that we identified as being transcriptionally regulated by LPS or IL-4 stimulation. Since the number of kinases transcriptionally regulated by the two stimuli is very high, in this case we chose to apply a more stringent cutoff (>0.5), eventually resulting in a total of 58 inhibitors screened.

Importantly, the deconvolution process used in our study heavily relies on the redundancy and overlapping effects of these inhibitors. The decision to avoid excessive stringency was motivated by the necessity to ensure a sufficient number of inhibitors, allowing for redundancy on the one hand and comprehensive coverage of pathways on the other. A more stringent approach might have led us to overlook critical similarities due to an insufficient number of inhibitors.

While acknowledging this as a limitation, it also serves as an inherent advantage of the proposed screening method as relationships among inhibitors could in principle be constructed blindly without knowing the designated targets.

Relevant modification to the text can be found at rows 186-201.

“Why just one antibody for ChIP-seq?”

Indeed, multiple antibodies could have been used but as explained above this would have limited the feasibility of the study because of the limited throughput of ChIP-seq. Besides feasibility issues related to the number of samples analyzed, H3K27ac has been extensively characterized over 15 years of work as a histone modification pervasively associated with active *cis*-regulatory elements, including promoters and enhancers. Data from the literature show that (with specific exceptions, such as H4K16ac) most histone acetylation changes occur similarly on the same tails, which makes the analysis of different acetylated residues redundant in most cases.

To highlight how general H3K27ac marks is, we show below the correlation between the H3K27ac (ab4729) signal in resting and LPS treated macrophage to the one measured with the tetra-acetylated histone H4 (cat. no. o6866, Millipore) as recently published by our lab (Russo et al. Molecular Cell 2023, GEO Series GSE245444).

Figure 2. Correlation between H3K27ac (Abcam cat. no. ab4729) and tetra-acetylated histone H4 (Millipore cat. no. o6866). Signal was registered across all ~59,000 accessible sites. Read counts per kb were related considering either untreated conditions (left) or LPS-stimulated (2h) conditions (right). Spearman r was calculated and reported.

Methylation based marks are instead characterized by a more restricted distribution (e.g., preferential association of H3K4me3 with promoters) and/or a much more limited dynamic behavior (e.g., H3K4me1) in the context of short term stimulations and were thus not suitable for our approach.

"Please be more specific here and elsewhere: "by generating several hundred H3K27ac ChIP-seq data sets" How many and how do these come about?"

We have modified the text to be more specific and clearer regarding the number of samples generated. The relevant breakdown of the number of samples was nevertheless reported in the Materials and Methods section and also in the summary of the experimental design in Figure 1A and at the GEO repository GSE219240. The experimental design adopted resulted in ~600 ChIP-seq samples (corresponding to a full-time course per condition per CKI) in addition to the 18 samples for the DMSO Vehicle. This is now stated in the main text for clarity and can be found at row 230.

"MFA analysis: this text is rather inaccessible to a common reader and I struggled in many places to follow the descriptions/arguments. Please simplify, use clearer and more consistent language and focus the text on what the MFA analysis shows with regard to the purpose of the experiment."

We thank the reviewer for raising this critical issue. We do recognize the high complexity of the original text and the need to simplify the presentation of the Multiple Factor Analysis (MFA). We have now modified the relevant part of the text (see row 232 – 268 of the revised manuscript) in order to deliver the general concept behind the MFA, the reasons why it was employed and how we used it.

The details regarding the analysis are still extensively covered in the Materials and Methods section for specialized readers.

"I am surprised by the statement that the shift in the barycenter is a quantitative measure. It is not in PCA, so how come it is in MFA?"

We appreciate the reviewer's valuable insight and acknowledge the necessity for a clearer interpretation of our MFA analysis in the original text. In response, we have made revisions to enhance clarity, as detailed in rows 232-268.

The shift in the barycenter within the MFA averages the collective impact of all CKIs tested on individual CREs. Because this shift reflects the combined action of multiple CKIs, it serves as a quantitative measure. Hence, in contrast to PCA, where the barycenter lacks a meaningful interpretation due to the absence of grouping information in the principal component space, any shift in the barycenter in MFA holds significance.

Furthermore, the distances between CKIs and DMSO for each CRE represent CKI-specific perturbations on individual CREs. Being calculated across relevant dimensions (see row 282), these distances offer a quantitative measure of similarity based on the covariance or variance explained. By selecting components with variance greater than that from randomly permuted data, we effectively filter out noise, retaining the most meaningful variation. This process yields quantitative measures of similarity based on informative components. Both PCA and MFA preserve similarity, with Euclidean distances in the reduced-dimensional space directly reflecting distances in the original feature space (see rows 286-288), thereby capturing the underlying data structure. This approach enhances focus on relevant patterns while mitigating the influence of noise.

While we have simplified the main text for the sake of clarity, detailed analysis procedures are extensively covered in the Materials and Methods section for specialized readers.

"Does the shift in barycenter correlate with the number of targets of the CKI used? If yes, how can one attribute the effects to a particular kinase?"

As mentioned in the previous answer the CREs' barycenter is the average position considering all the conditions examined. Therefore, changes in position relative to the one obtained considering solely the DMSO condition, depend on the number of CKIs impacting H3K27ac at the same CRE, not on the number of targets that each inhibitor has.

In addition, our approach relies on the identification of CKIs' effects on the epigenome; not on the attribution of certain effects to a specific kinase perturbed by a given inhibitor. Hence, we cannot (and never claimed that we can) assign a given effect to a particular kinase. The pattern of H3K27ac at affected CREs are used as signatures of the effects caused by the CKIs, which depend on both the polypharmacological nature of the compounds, and the underlying set of kinases active in the cells when stimulated.

The CKIs' specific effects, as extensively described in the Materials and Methods and now re-phrased in the main text (row 261-281), are computed per CRE by considering the distance between the DMSO condition and the individual inhibitors therefore obtaining a measure of the effects related to the specific inhibitors as a function of all measured distances.

"What is an MFA dimension? Why was it necessary to compute a perturbation likelihood when your assay is supposed to measure perturbation directly?"

We thank the reviewer for providing the opportunity to clarify these points. We have revised the text accordingly to improve clarity.

An MFA dimension represents a latent variable that captures the variability in the dataset attributed to specific factors or conditions. These dimensions are constructed based on linear combinations of the original variables, allowing to examine how different factors contribute to the observed variation in the data.

As depicted in Figure 1B jointly with 1D, the first dimensions captured about 70% of the variability within the dataset and reflect the kinetics of activation of individual CREs. This relationship is now more explicitly stated in the text at row 250-260. Supplementary Figure S2F-G describes in detail the relationship between components and kinetics.

As for the necessity of computing a perturbation likelihood, this was done to derive a single metric summarizing the differences from the control condition across the full time-course. This approach was necessary due to the constraints of performing a single time-course per perturbation and the need for a straightforward method to quantify changes without explicit kinetic modeling. Despite the single-time-course screening approach for CKIs, where the CREs changing upon LPS or IL4 were selected using repeated measurements of the time-course in the control DMSO condition, this evaluation was cautiously conducted considering our large number of pairwise distances, repeated time points per perturbation, pre-selection of differential CREs based on the control condition, along with normality testing and good fit to Gaussian distribution (see Appendix Fig S3A).

We appreciate the reviewer's input, which has contributed to a clearer understanding of our methodology and its rationale.

"It is unclear to me what the purpose of the 2D network-graph is."

The 2D network graph serves as a visual representation that overcomes the limitations imposed by the mono-dimensionality of dendrograms and hierarchical clustering. Unlike dendrograms, where elements are positioned in a hierarchical structure and can be close to only two other elements at most, the 2D network graph provides a more flexible and unbiased visualization. This flexibility is particularly important because dendrograms can exhibit biases depending on the specific hierarchical clustering method employed. The 2D network graph overcomes this limitation by allowing elements to be positioned more freely still through optimization based on real Euclidian distances, providing a clearer and less method-dependent representation of relationships among elements.

"And I do not see a justification for the conclusion that "...effects of individual CKIs are influenced by the underlying network of active signaling pathways...". It is much more likely that the lack of such correlation is rooted in the polypharmacology of the drug (i.e. more than one potent target). This is a fundamental issue in this analysis at least in the way the authors are trying to portray it. Up to this point it is not at all clear how the generated data can be meaningfully interpreted."

We appreciate the reviewer's observation and we apologize for the lack of clarity on our side. The statement in question related to our interpretation of the observed dissimilarities of distances among CKIs observed in the LPS and IL-4 conditions. We surmise that since all the CKIs used target multiple kinases, the nearest neighbor pattern changed when cells were activated by two different stimuli just because the signaling pathways were differentially rewired by the two stimuli. For instance, Momelotinib targets both TBK1 and JAK kinases. TBK1, however, is activated only by LPS and not by IL-4, while JAKs are activated by both LPS and IL-4. Hence, Momelotinib will perturb different sets of CREs when macrophages are activated with LPS or IL4 and its relationship to other inhibitors targeting TBK1 or JAKs will be different in cells stimulated with LPS and in cells stimulated with IL-4. We have now amended the pertinent text (see row 303-308).

"# Section on "Proximity among CKIs as a function of kinetics and genomic overlap of H3K27ac changes." I could repeat many of the comments above. Please find a way to state the purpose of the analysis and formulate the results in a way that can be interpreted. It starts with the title of the section: what is "proximity among CKIs"? From my read of the text, it is entirely unclear how the authors arrive at this conclusion: "Altogether, these observations suggest that CKIs proximity based on H3K27ac changes, reflects the global spectrum of actions of the individual inhibitors." At no point in this section do the authors state what the action of these inhibitors actually is. What are the transcriptional programs they do or do not affect? Do these bear any relationship with known cellular consequences of these drugs in the chosen biological system?"

We do recognize the confusion caused by the way certain results were presented and we have made extended amendments to the relevant part of the text (see 313 onwards).

In more details, this section now states explicitly the aim, which is to interpret what we obtained from the analysis of CKI similarities based on the H3K27ac changes (row 316-317)

As regards "proximity", we referred to the similarities among CKIs based on H3K27ac, and we agree with the reviewer that the notion of "proximity" may be difficult to understand and not intuitive at all. We have modified the sub-section title and replaced proximity with similarities in terms of H3K27ac changes.

The step-by-step analysis is now extensively being re-written in order to better display the logic and take-home messages from each analysis (row 315-330 and 338-352).

By stating that "Our assumption was that the kinetics of CKI-induced changes in stimulus-regulated H3K27ac could offer insights into the signaling pathways and downstream TFs they inhibit" we clearly evaluated two distinct concepts 1) whether CKIs effects have a temporal bias 2) the extent of overlapping effects between pair of CKIs. The high level of overlap and the shared bias in time between similar CKIs (still using our read-out) might well be the consequence of impairing common sets of kinases.

Regarding the term 'action', we concur with the reviewer that it may lead to confusion. Unlike assays focusing on defining the 'action' of compounds by identifying their targets, our assay does not provide direct target-related information. We changed the term "action" with transcriptional alterations (row 373-375).

We believe that the observation of unrelated CKIs leading to similar genomics effects in a context of active signaling like the one triggered by LPS in macrophage is per-se a strong indication that are causing the impairment of shared sets of kinases.

We thank the reviewer for having highlighted these aspects. We now think that through these valuable insights we actually improved the text improving the impact of our findings.

"# Section on "Deconvolution of CKI effects by identification of CKI-regulated transcription factors.": This reviewer failed to gather what the authors are trying to achieve in this section and the results of the ROC deeply concern me. Also, Midostaurin is among the least selective CKIs we know today. So how can one envisage using an epigenetic profile to predict what the drug is going to do? If this is not the intent of the analysis, this reviewer must be missing an important point."

We understand this reviewer is concerned by the high-quality predictive models we generated for inhibitors with poor specificity, in particular Midostaurin.

Our predictive models, constructed using XGBoost, underwent thorough evaluation to ensure stability and effectiveness. Given the imbalanced nature of the dataset, we leveraged techniques such as repeated stratified k-fold cross-validation and class weighting to address potential biases and instability. Moreover, we were mindful of the potential for overfitting and over-optimistic ROC scores, hence we included Precision-Recall (PR) values to provide a more comprehensive evaluation.

It's important to note that while our models demonstrate effectiveness in predicting down-regulatory effects of CKIs on transcription factors (TFs), their performance in predicting up-regulatory effects is less robust.

Regarding Midostaurin, which is acknowledged as a poor specificity CKI, our model's strength lies in its capacity to predict epigenomic effects based on many diverse TF ChIP-seq datasets. The model demonstrates precision in forecasting down-regulatory effects, which holds true not only for

Midostaurin but also for other compounds having equally more targets. This suggests, as reported in the text, that despite the polypharmacological nature of these compounds, the underlying network is crucial. It's essential to highlight that our assay involves signal-induced changes rather than reporting changes retrieved from cells in resting conditions.

This section of the manuscript has now been revised for the sake of clarity (see row 396 onward).

"# Section on "Distinctive and unique properties of individual JAK inhibitors (JAKi)": This section of the manuscript was easier to follow but the results are trivial and the interpretation obvious. The fact that these JAK inhibitors have other targets, sometimes potent, should place them in all kinds of areas of the CKI effect space. What is more relevant though is if the biological system has active signaling components that utilize these targets. This section highlights an issue in the overall approach in that the authors used in-lysate kinobead data and poor selectivity compounds to try and design a cellular perturbation experiment. One cannot assume that the potency of kinobead binding translations into a similar potency in cells. This will have to be assessed on a case by case basis and may well explain why some of the JAK inhibitors did not do anything to STAT phosphorylation in the flow assay at the presumed EC50 from kinobeads. Just a not, the kinobead data reports apparent dissociation constants, not EC50 values."

We appreciate the reviewer's comments and have made revisions to enhance clarity (see row 508-513 and 530-541).

First, it's important to clarify that our analysis does not dispute the classification of these compounds as JAK inhibitors (JAKi). We have reworded our explanation to explicitly state that the observed reductions in CREs associated with IRF/STAT transcription factors, alongside with the down-regulation of genes controlled by IRF/STAT, indicate their inhibitory effects on JAKs. Notably, our epigenome-based readouts also outperformed p-STAT1-based assays.

Secondly, while variations in the concentrations used may influence the strength of downstream effects, our key observation remains unchanged: these compounds exhibit JAK inhibitory effects, regardless of their classification by kinobeads assay. This highlights a limitation of kinobeads in fully capturing the spectrum of a compound's inhibitory actions. Furthermore, our findings suggest that changes in inhibitor concentrations proportionally affect their engagement with potential targets. Higher concentrations may lead to increased JAK inhibition but also result in more pronounced effects on other targets.

To validate our observations, we conducted experiments in human monocyte-like cells using different stimuli. These validations confirmed the selective JAK inhibition of Filgotinib, identified Midostaurin as a selective TBK1 inhibitor, and revealed Momelotinib as a dual inhibitor of TBK1 and JAK. This cross-validation underscores the robustness of our findings across different experimental setups (see row 622-634).

In summary, although we didn't determine dissociation constants for each compound against specific targets, our study demonstrates that despite variations in concentrations, the observed genomic effects of these compounds can be rationalized within the context of intact and active signaling pathways. This surpasses the limitations of kinobeads data and other assays.

Additionally, regarding the definition of EC50, we amended the text to clarify that we used the deduced EC50 from the kinobeads assay as reported in the supplementary Table S2 (sheet "kinobeads") of Klaeger et al., 2017. This involved determining the concentration at the inflection point of the fitted curve where half of the kinases/beads association is lost.

"# Section on "Unexpected effects of CKIs on the IFN β -regulated gene expression program." The results of the GO analysis are obvious insofar as macrophages are cells of the immune system and changes in their epigenomes/transcriptional profiles will always produce strong GO terms related to the basic underlying biology. The same would be observed in eg brain or liver cells. This is a well known limitation of current GOs and cannot be taken as evidence the way suggested by the authors".

Actually, the GO analysis was a marginal and in fact unnecessary addition to this section and the narrative has been changed accordingly. Here, our main point was that some inhibitors increased IFN β gene expression and the downstream IFN program (which is a highly specific one with a very regulated activation pathway).

"There is a misconception in this paragraph regarding the CETSA assay. Although widely portrayed as such, drug binding does not necessarily increase the thermal stability of its target. The number of cases that do the opposite is just as high and many drugs do not show any thermal shift. CETSA shifts just record differences in the energy landscape of/around the protein that may be a direct or indirect consequence of drug binding. But yes, the results of the authors imply that their assay can be more informative than a CETSA."

We thank the reviewer for highlighting this point. We have now amended the text to better explain known issues with the interpretation of CETSA assay both in the introduction and in the results section (see row 98-101 and row 609).

Reviewer #2:

"Gualdrini et al present an innovative study with massive amounts of data, profiling the epigenomic effects of 58 clinically approved kinase inhibitors. Each inhibitor is tested in affecting the LPS- and/or IL4-response of macrophages, characterized by a short timecourse of H3K27Ac, RNA-seq data at the 2 hr timepoint, and finally ChIP-seq studies of dozens of stimulus-induced transcription factors. This is a uniquely comprehensive dataset that has the potential to yield novel insights.

The dataset also poses some data analysis challenges. The manuscript details the rationale of the data analysis workflow and the description seems comprehensible to a fairly general/informed audience and makes sense. At the same time there is no doubt that alternative analysis workflows are possible, and these may lead to other insights. The present analysis is largely restricted to learning about the specificity or breadth of the pharmacological profile of each CKI, and comparing it to in vitro data. This makes sense and some insights of general interest are learned. (The dataset may also be of interest for other questions, for example from the perspective of predicting gene expression and gene regulatory mechanisms, with the CKIs being used as perturbations to develop a model). As a result there is also a wealth of granular information about the specificity or overlap of particular CKIs that may be of interest to small numbers of researchers. Some of these are worked through to gain mechanistic detail.

Overall this is an impressive and novel study that deserves to be published. While the insights hold some interest, the datasets have value in their own right as they allow many further studies. The manuscript is well written. Here some minor points:

58 CKI were chosen: why are they described as non-overlapping: when some target the same kinase, and 238 pairs shared more than one target? This is quite remarkable and it would be nice to see this enumerated in the main text/figure."

We thank this reviewer for the enthusiastic appreciation of our study and for recognizing its potential value also for studies related to gene regulatory mechanisms.

As for the issue on the overlap among the 58 CKI uses, we do totally agree with the reviewer and apologize for the confusion. We have now removed the term "non-overlapping" and amended the text to better clarify the known target overlaps. Indeed, these 58 inhibitors exhibit a considerable range of overlap in terms of targets. This observation is also evident when examining Appendix Figure S1, where we provide CATDs for each target for each CKI. Actually, the deconvolution process we used in our study heavily relies on the redundancy and overlapping effects of these inhibitors.

p.4 homogeneous criteria => consistent criteria

Thanks for the suggestion.

IL4 - not sure that referring to it as a homeostatic stimulus makes sense. It is a microenvironmental cytokine that also elicits macrophage responses.

We do agree and have now amended the relevant part of the text (row 133-135).

p.8 Defined as the size of the intersection of the CREs affected by two CREs => CKIs

Thanks for noticing it.

Reviewer #3:

"This manuscript (An integrative epigenome-based strategy for unbiased functional profiling of clinical kinase inhibitors) from the Natoli lab presents an integrative analytical approach to identify similarities and differences among clinical kinase inhibitors (CKIs) with varying specificities, based on chromatin modifications (H3K27ac). Specifically, the authors use the well known model of inflammatory responses in bone marrow derived macrophages stimulated either with LPS or IL4 for different time frames.

This is an interesting, novel systems biology approach to compare the effects of the different compounds on a global scale, at the genomic/transcriptional level. In general, the data is well presented and of interest to a broad audience. I would therefore recommend publication in Mol Syst Biol, once the following comments have been addressed:

- The authors observed that H3K27ac marks had a substantial degree of overlap in specificities among CKIs, but no such overlaps were seen in the RNAseq computed model (Figure 2). (Could this be due to differences in treatment/time points?) Moreover, several inhibitors unexpectedly maintained the activation of immune response upon LPS stimulation (Figure 5). Could the authors please discuss and elaborate on their interpretation."

We thank the reviewer for recognizing the novelty and relevance of our approach while at the same time providing constructive suggestions for improvement.

We appreciate the specific comment provided by the reviewer. Although the entire analysis on H3K27ac was conducted in a time-resolved manner, the comparison between H3K27ac and RNA-seq readouts in Figure 2 (panels G and H) specifically focused on the 2-hour time point. This decision was made for the sake of consistency, as the RNA-seq screen was exclusively performed at this particular time point (see row 383).

What we conveyed in our report is that, regardless of the time points considered, the number of changes captured by RNA-seq is lower. Several factors may contribute to this discrepancy, including the impact of RNA stability on RNA abundance, the magnitude of changes, and, possibly more importantly, the sheer abundance of data points (which is clearly greater for H3K27ac). In summary, the RNA-seq readout might be influenced by mechanisms that extend beyond the simple activation or perturbation of upstream signaling events, thereby limiting its sensitivity.

The mechanisms associated with the activation of *Ifnb1* observed for Omipalisib, BGT-226, Mubritinib, Lapatinib, IMD-0354, Lenvatinib, and Tofacinib appeared to be CKI-specific. Despite efforts to identify a common target using kinobeads or designated targets, these attempts proved unfruitful. MS-CETSA assays conducted in intact macrophages also failed to reveal any commonality among these inhibitors (these data were not included in the manuscript because they are inconclusive). Consequently, we explored whether the activation of *Ifnb1* was linked to viral mimicry mechanisms associated with the dysregulation of Transposable Elements (TE), as reported in Figure 5. While an increase in transposable elements was detected, this finding does not fully clarify the specific actions of these inhibitors in this context.

"- I may have missed or overlooked this part, but did the authors check and present CKIs that had no effects, i.e. that did not have effects on H3K27ac levels under the conditions tested?"

Given the polypharmacological nature of CKIs, finding inhibitors with no-effects whatsoever on signal transduction in either LPS or IL-4 conditions is highly unlikely. This complexity is evident in our analysis of inhibitors like BMS-911543, which, based on kinobeads assay should selectively target Nqo1 but in fact caused unexpected effects on H3K27ac and transcriptional changes. Another instance is Filgotinib, which is predicted by kinobeads to have no impact on Jak kinases, yet is a potent inhibitor of STAT1 phosphorylation. The limitations of existing technologies for defining target specificity make the definition of a ground truth for each inhibitor difficult or even impossible.

We leveraged the extensive inhibitor screening and the low doses employed (theoretical or apparent dissociation constant computed in cell lysate from kinobeads assay) to re-assess the impact of these CKIs in the specific context used. The large number of inhibitors and their low dosages allowed us to evaluate their impact on histone acetylation changes, offering a comprehensive perspective. In principle, one could blindly assess the impact of any compound, regardless of its designated target, on histone acetylation changes and compare its effects with these screened compounds.

"- Also, in general, H3K27ac patterns and TF activities are lineage and cell type dependent. The findings presented here are an excellent example of innate immune responses in murine macrophages, but of course, validation in human cells or in other cell types would be required to generalize the conclusions drawn here."

We do agree with the reviewer's comment regarding the lineage and cell type-dependent nature of H3K27ac patterns and TF activities. To address this concern and enhance the study's robustness, we conducted targeted validations in additional contexts. Specifically, we validated the extracted observations regarding Midostaurin as a potent TBK1 inhibitor in human monocyte-like THP1. The validation was conducted by including also Filgotinib (the most potent JAK inhibitor in our assay) and Momelotinib (potentially both a TBK1 and a JAK inhibitor from our assay) and assessing *IFNB1* and *RSAD2* gene expression following LPS, dsDNA or IFN β stimulation. The data is now presented in **Figure S7** (panel G and H) and the relevant results presented at row 622 onward.

"- The selection criteria for the kinase inhibitors included their targets being components of the TLR4/IL-4R signaling pathways, as well as the kinases being transcriptionally activated in response to LPS or IL-4. Obviously, more kinases or rather CKIs could be added to the list, given that many of them may be regulated at the post-translational level (protein expression, PTMs). This point may be emphasized in the discussion section, if it is not feasible to add more data."

We appreciate the reviewer's insightful observation. Indeed, the inclusion of more kinase inhibitors, potentially regulated at the post-translational level, as well as the exploration of diverse cell types and stimuli, could enhance the comprehensiveness of our study. While it may be feasible to expand the scope of the assay, we would like to emphasize that the focus of our presented study was on introducing a complementary assay with a readout serving as a proxy for signaling perturbation and integration. This readout, we believe, provides a more apical and interpretable measure compared to general and more complex readouts such as cell growth or motility.

In response to the reviewer's suggestion, we have incorporated this discussion in the manuscript's relevant section, underscoring the potential for expanding our assay to encompass a broader range of compounds and conditions. We are grateful for this constructive feedback, and we believe that this enhancement strengthens the broader applicability and future prospects of our presented assay.

"Minor comments:

- page 5, paragraph 1; the last line appears to be an incomplete sentence (after "and the PI3 kinase")."

We apologize for this error. The sentence has been completed and corrected and now can be read as:

"For instance, Bafetinib, an approved BCR-ABL and Lyn tyrosine kinase inhibitor (EC_{50} for ABL1 = 26,37 nM), was used here as a MAPKAPK3 inhibitor (EC_{50} = 23,1 nM) and the PI3 kinase inhibitor Apitolisib was used to target JAK1 (EC_{50} = 25,8 nM) (Appendix Fig S1 and Dataset EV1)."

"- page 5, paragraph 3, line 2-3; it's mentioned "and about half of them could be assigned to a total of 208 CKIs with a wide range of selectivity scores". However, Figure 1 illustrates 177 CKIs in total. Please correct this inconsistency."

In response to the reviewer's query regarding the apparent discrepancy between the text on page 5, paragraph 3, lines 2-3, and Figure 1, we would like to clarify that the figures and numbers presented in Figure 1A specifically pertain to components within the respective signaling pathways. For the TRL4 pathway, there are 127 components, of which 35 are kinases; for the IL4 pathway, there are 112 components, with 6 being kinases. Within these, 25 druggable components in the TRL4 pathway and 5 in the IL4 pathway, each associated with at most 2 CKIs from the kinobeads data (with a CATDS threshold of 0.1), were identified. Moving to section "2," the figures and numbers also relate to the kinases, not the CKIs. There are 234 and 74 kinases that exhibit changes in expression levels upon LPS or IL4 stimulation, respectively. When stating that 'about half' of these kinases could be targeted, we refer to 117 out of 234 for LPS and 37 out of 74 for IL4, by a total of 208 CKIs. Therefore, it is crucial to note that the number 208 pertains to the CKIs targeting the identified kinases, not the kinases themselves.

"- Page 5, paragraph 3, line 4-5; the text mentioned that "with a CATDS > 0.5, eventually resulting in a list of 24 inhibitors". However, Figure 1 shows 23 inhibitors. Please correct."

In addressing the concern raised by the reviewer regarding the apparent inconsistency between the text on page 5, paragraph 3, lines 4-5, we would like to clarify that the figures and numbers refers to the selection of inhibitors based on the identified kinases. To provide further clarification, the numbers 17 out of 117 and 6 out of 37, mentioned previously, relate to the kinases, not the inhibitors. The 23 kinases illustrated in Figure 1 are utilized for the selection of inhibitors, and in this case, we indeed have 24 inhibitors.

In essence, Figure 1A, aside from presenting the schematic of individual signaling pathways and the experimental flow, primarily depicts the selection of CKIs based on kinases. The sections labeled 'TKR4/IL4R signaling components' and 'Stimulus-regulated components' report the number of kinases, not CKIs. Conversely, the Venn diagram on the right side of the figure indicates the number of CKIs selected according to the identified kinase targets.

8th Apr 2024

Manuscript Number: MSB-2023-12088R

Title: An integrative epigenome-based strategy for unbiased profiling of clinical kinase inhibitors

Dear Dr. Natoli,

Thank you again for submitting your work to Molecular Systems Biology. We have now heard back from two of the original reviewers who evaluated your study. As you will see below, although the reviewers seem to be mostly satisfied with the revisions, there are still important concerns that have not been addressed, and therefore we would ask you to further revise your manuscript. In particular, Reviewer 1 does not find the presentation suitable and comments that there are overinterpretations that should be toned down and alternative explanations should be offered. Importantly they also point out that there are discrepancies between the point-by-point response letter and the actual changes made in the manuscript. These points will need to be addressed in full in order to proceed with your manuscript.

As in the previous decision letter, I am copying the instructions for formatting, etc below:

1) A .docx formatted version of the manuscript text (including legends for main figures, EV figures and tables). Please make sure that the changes are highlighted to be clearly visible. Alternatively you may choose to submit your manuscript as a LaTeX file.

4) A .docx formatted letter INCLUDING the reviewers' reports and your detailed point-by-point responses to their comments. As part of the EMBO Press transparent editorial process, the point-by-point response is part of the Peer Review File (PRF), which will be published alongside your paper.

5) A complete author checklist, which you can download from our author guidelines (<https://www.embopress.org/page/journal/17574684/authorguide#submissionofrevisions>). Please insert information in the checklist that is also reflected in the manuscript. The completed author checklist will also be part of the PRF.

6) Please note that all corresponding authors are required to supply an ORCID ID for their name upon submission of a revised manuscript.

7) It is mandatory to include a 'Data Availability' section after the Materials and Methods. Before submitting your revision, primary datasets produced in this study need to be deposited in an appropriate public database, and the accession numbers and database listed under 'Data Availability'. Please remember to provide a reviewer password if the datasets are not yet public (see <https://www.embopress.org/page/journal/17574684/authorguide#dataavailability>).

This study includes no data deposited in external repositories.

8) For data quantification: please specify the name of the statistical test used to generate error bars and P values, the number (n) of independent experiments (specify technical or biological replicates) underlying each data point and the test used to calculate p-values in each figure legend. The figure legends should contain a basic description of n, P and the test applied. Graphs must include a description of the bars and the error bars (s.d., s.e.m.). Please provide exact p values.

10) We replaced Supplementary Information with Expanded View (EV) Figures and Tables that are collapsible/expandable online. A maximum of 5 EV Figures can be typeset. EV Figures should be cited as 'Figure EV1, Figure EV2' etc... in the text and

their respective legends should be included in the main text after the legends of regular figures.

<https://www.embopress.org/page/journal/17574684/authorguide#expandedview>

11) For more information: There is space at the end of each article to list relevant web links for further consultation by our readers. Could you identify some relevant ones and provide such information as well? Some examples are patient associations, relevant databases, OMIM/proteins/genes links, author's websites, etc...

12) Author contributions: CRediT has replaced the traditional author contributions section because it offers a systematic machine readable author contributions format that allows for more effective research assessment. Please remove the Authors Contributions from the manuscript and use the free text boxes beneath each contributing author's name in our system to add specific details on the author's contribution. More information is available in our guide to authors.

13) Disclosure statement and competing interests: We updated our journal's competing interests policy in January 2022 and request authors to consider both actual and perceived competing interests. Please review the policy <https://www.embopress.org/competing-interests> and update your competing interests if necessary.

14) Every published paper now includes a 'Synopsis' to further enhance discoverability. Synopses are displayed on the journal webpage and are freely accessible to all readers. They include a short stand first (maximum of 300 characters, including space) as well as 2-5 one-sentences bullet points that summarizes the paper. Please write the bullet points to summarize the key NEW findings. They should be designed to be complementary to the abstract - i.e. not repeat the same text. We encourage inclusion of key acronyms and quantitative information (maximum of 30 words / bullet point). Please use the passive voice. Please attach these in a separate file or send them by email, we will incorporate them accordingly.

Share synopsis text and image, as well as eTOC:

Please note that these would be the final versions and changes during proofing are usually not allowed

15) As part of the EMBO Publications transparent editorial process initiative (see our policy here:

https://www.embopress.org/transparent-process#Review_Process), Molecular Systems Biology will publish online a Peer Review File (PRF) to accompany accepted manuscripts.

In the event of acceptance, this file will be published in conjunction with your paper and will include the anonymous referee reports, your point-by-point response and all pertinent correspondence relating to the manuscript. Let us know whether you agree with the publication of the PRF and as here, if you want to remove or not any figures from it prior to publication.

Please note that the Authors checklist will be published at the end of the PRF.

Molecular Systems Biology has a "scooping protection" policy, whereby similar findings that are published by others during review or revision are not a criterion for rejection. Should you decide to submit a revised version, I do ask that you get in touch after three months if you have not completed it, to update us on the status.

Click on the link below to submit your revised paper.

Yours sincerely,

Poonam Bheda, PhD
Scientific Editor
Molecular Systems Biology

Reviewer #1:

The revised manuscript has been improved in several places.

As noted by another reviewer, the strength of the work is an interesting assay and a large body of data. The clear weakness still is how the work is presented and conclusions drawn. Much of the text is still inaccessible to readers with a biological background and it still remains unclear in most places what one actually learns from the data. The part dealing with 'predictions' clearly is overinterpreted. And in many other cases the reader has to 'believe' the interpretation provided of the authors while this reviewer could, in most cases, also offer alternative explanations that would be equally well/poorly supported by the data. This reviewer also notes that there are discrepancies between the the point to point response and changes made in the actual manuscript.

Reviewer #3:

In this revised version, the authors have addressed (most of) my points raised, I have nothing further to add.

MSB-2023-12088 – Gualdrini et al. *An integrative epigenome-based strategy for unbiased functional profiling of clinical kinase inhibitors*

We once again thank the reviewers for the time dedicated to assess and improve our work. In response to the Reviewer #1's comments, we have modified the manuscript in order to conceptualize the rationale and the outcomes of the methodological approaches while moving all technicalities to the Material and Methods section. This has clearly resulted in much improved readability without impacting the content of our study.

Moreover, we have meticulously evaluated each statement and conclusion, rephrasing some sentences or indicating, when appropriate, possible alternative explanations of our findings.

Finally, we do apologize for the inconsistencies between the previous point-by-point letter and the manuscript text. Such discrepancies were only and exclusively the consequence of formatting issues generated during the online conversion of the word file uploaded onto the system website.

Reviewer #1:

"The revised manuscript has been improved in several places.

As noted by another reviewer, the strength of the work is an interesting assay and a large body of data. The clear weakness still is how the work is presented and conclusions drawn. Much of the text is still inaccessible to readers with a biological background and it still remains unclear in most places what one actually learns from the data. The part dealing with 'predictions' clearly is overinterpreted. And in many other cases the reader has to 'believe' the interpretation provided of the authors while this reviewer could, in most cases, also offer alternative explanations that would be equally well/poorly supported by the data.

This reviewer also notes that there are discrepancies between the the point to point response and changes made in the actual manuscript."

We express our gratitude to this reviewer for encouraging us to enhance the clarity of our work, which we agree is a critical issue.

We have now streamlined the presentation of our analysis, particularly regarding the core MFA method used in the section "*Histone acetylation as unbiased and specific readout of CKIs' effects*". A paragraph has been added to elucidate the rationale behind employing this method and the aims we intended to pursue.

We have also simplified the presentation by either removing or simplifying technical details, which are however comprehensively explained in the Materials and Methods section.

Additionally, in response to this reviewer's suggestion, we have scrutinized the individual conclusions drawn from our analysis. Specifically, we have underscored that the polypharmacological nature of the various compounds utilized is the primary confounder affecting the correlation between CKIs with identical or similar targets. These amendments have been included in the "*Histone acetylation as unbiased and specific readout of CKIs' effects*" section.

We have also revised our interpretation regarding correlated H3K27ac changes caused by unrelated CKIs, stating that they may result from the targeting of kinases that converge on the same signaling pathways and branches. Previously, we hypothesized that these effects may stem from multiple CKIs targeting the same kinases, but we concur with the reviewer that although this is the most parsimonious hypothesis, there is insufficient evidence to conclusively and unambiguously demonstrate it. These revisions are highlighted in the third paragraph of the section "*Similarities among CKIs as a function of kinetics and genomic overlap of H3K27ac changes*".

Furthermore, we have refined and strengthened the conclusions regarding the relationship between H3K27ac and transcriptional changes. We now emphasize that H3K27ac changes can capture both

correlative and anti-correlative transcriptional alterations, as outlined in the section "*Similarities among CKIs as a function of kinetics and genomic overlap of H3K27ac changes*".

The section pertaining to the machine learning models generated to predict CKI-induced H3K27ac changes based on genome-wide TF occupancy has been extensively revised for clarity and made fully understandable to scientists without a specific quantitative background. The presentation and objectives of this section have been simplified, with a reduction in the complexity of the text concerning the machine learning strategy and feature extraction method.

Additionally, the conclusion of this section now highlights the challenges of disentangling what combination of inhibited kinases leads to H3K27ac changes. Indeed, we can correlate H3K27ac alterations vs. sets of TFs rather than directly linking specific kinases to the regulation of specific TFs. These modifications are reported in the "*Deciphering the Mechanistic Impact of CKIs on the Epigenome through TF Regulation*" section.

Lastly, we reiterate that the interpretation of the relationship between the JAK inhibitors is complicated by the polypharmacological nature of these compounds. This revised perspective is presented in the "*Distinctive and unique properties of individual JAK inhibitors (JAKi)*" section.

Regarding the discrepancies between the point-to-point response and changes made in the actual manuscript, we apologize for this inconvenience which related, however, only to the lines indicated in the letter. This issues occurred because of a change in manuscript formatting during the online conversion of the word file we uploaded in the system. Nevertheless, the individual sections indicated in the letter were all highlighted in yellow within the text and hence were easily identifiable.

Below, we outline the discrepancies found, referencing the current row numbers:

- The additional introduction concerning the literature on phosphoproteomics should now be referred to rows 101-109.
- Changes from "unbiased functional information readout" to "specific" can be found at row 112.
- Details regarding the CATDs metrics in relation to the concentration used are located at rows 209-216.
- The rationale for using a CATDs score with a lower limit of 0.1 is provided at rows 187-190.
- Specific explanations regarding the number of generated datasets (approximately 600 Chip-seq datasets) can be found at row 228.
- Modifications to the text related to the MFA methodologies can be found starting at row 231. The interpretation of the relative shift in the position of individual CRE per CKI within the MFA compromise can be found at rows 260 and 271 of the latest version.
- Amendments made to better explain what an MFA dimension is now present at rows 253-259.
- Changes to the conclusions drawn from the first result section are now located at rows 293-299.
- Changes to the second result sections begin from row 306 onwards.

15th Apr 2024

Manuscript Number: MSB-2023-12088RR

Title: An integrative epigenome-based strategy for unbiased profiling of clinical kinase inhibitors

Dear Dr. Natoli,

Thank you for the submission of your revised manuscript to Molecular Systems Biology. I am pleased to inform you that we will be able to accept your manuscript pending the following final amendments:

- 1) In the main manuscript file, please reduce keywords to max. 5.
- 2) Please format the Data availability section describing how the data, code etc. have been made available according to the example below:
"The datasets and computer code produced in this study are available in the following databases:
- Chip-Seq data: Gene Expression Omnibus GSE46748 (<https://www.ncbi.nlm.nih.gov/geo/query/acc.cgi?acc=GSE46748>)
- Modeling computer scripts: GitHub (<https://github.com/SysBioChalmers/GECKO/releases/tag/v1.0>)
- [data type]: [full name of the resource] [accession number/identifier] ([doi or URL or identifiers.org/DATABASE:ACCESSION])"
- 3) Please ensure that the sequencing dataset GSE219240 is now publicly available.
- 4) Please rename "Competing interests" to "Disclosure and competing interests statement". We updated our journal's competing interests policy in January 2022 and request authors to consider both actual and perceived competing interests. Please review the policy <https://www.embopress.org/competing-interests> and update your competing interests if necessary.
- 5) Author contributions: Please remove it from the manuscript and specify author contributions in our submission system. CRediT has replaced the traditional author contributions section because it offers a systematic machine-readable author contributions format that allows for more effective research assessment. You are encouraged to use the free text boxes beneath each contributing author's name to add specific details on the author's contribution. More information is available in our guide to authors:
<https://www.embopress.org/page/journal/17574684/authorguide#authorshipguidelines>
- 6) Please check the "Author Checklist" carefully and complete all relevant questions. Currently information about animals (mice) is missing in the "Experimental animals" section.
- 7) In the Materials and Methods, please take care of the following:
 - Please ensure that a statement on whether or not blinding was done is included in the Materials and Methods even if no blinding was done. Please also ensure this part of the Author Checklist is updated.
 - Animals: Please ensure that housing conditions as husbandry of the animals involved in experiments is reported. Please also ensure this part of the Author Checklist is updated.
 - Antibodies: Please ensure that dilutions/amounts of each antibody used are reported in the relevant Materials and Methods sections.
- 8) Please place individual sections of the manuscript in the following order: Title page - Abstract & Keywords - Introduction - Results - Discussion - Materials & Methods - Data Availability - Acknowledgements - Disclosure and Competing Interests Statement - References - Figure Legends - Expanded View Figure Legends.
- 9) For the figures and figure legends, please take care of the following:
 - Please define the annotated p-values ****/*** in the legend of figure 2h. Please note we require exact p-values to be reported.
 - Please indicate the statistical test used for data analysis in the legends of figures 2h; 4f-g; 5a, c-f; 6d.
 - Please note that the box plots need to be defined in terms of minima, maxima, centre, bounds of box and whiskers, and percentile in the legends of figures 2d, f, h; 6f.
 - Please note that information related to n is missing in the legends of figures 2d, h; 5c; 6e.
 - Please note that for heatmap present in figure 2b; a numbered scale bar is not provided. This needs to be rectified.
 - Expanded View Dataset legends should be removed from the main manuscript file and rather each one should be individually included as a separate tab in the relevant Excel file.
 - Appendix info ("Appendix/Appendix Figures S1 to S7") should be removed from main manuscript file.
- 10) Please ensure that all funding sources are entered into the manuscript submission system. "This work was also partially supported by the Italian Ministry of Health with the "Ricerca Corrente" and "5x1000" funds to the IEO IRCCS." is inserted in the comments box, but needs to be included in the "More Funders" list. A fellowship from the Associazione Italiana Ricerca sul Cancro (AIRC) is also missing from the manuscript submission system.
- 11) Synopsis:
 - Synopsis image: Please upload as a high-resolution jpeg file and ensure that the dimensions are 550 pixels wide x (250-400) pixels high.
 - Synopsis text: Please shorten the standfirst (maximum of 300 characters, including spaces) and limit it to 1 sentence.
 - Please check your synopsis text and image before submission with your revised manuscript. Please be aware that in the proof stage minor corrections only are allowed (e.g., typos).
- 12) Please upload the Appendix as a single PDF (no separate image files are needed).
- 13) Please ensure that a completed Source Data checklist (sent to you by Hannah Sonntag) is uploaded.

14) As part of the EMBO Publications transparent editorial process initiative (see our policy here: https://www.embopress.org/transparent-process#Review_Process), Molecular Systems Biology will publish online a Peer Review File (PRF) to accompany accepted manuscripts. This file will be published in conjunction with your paper and will include the anonymous referee reports, your point-by-point response and all pertinent correspondence relating to the manuscript. Let us know whether you agree with the publication of the PRF and as here, if you want to remove or not any figures from it prior to publication. Please note that the Authors checklist will be published at the end of the PRF.

15) Please provide a point-by-point letter INCLUDING my comments as well as the reviewer's reports and your detailed responses (as Word file).

I look forward to reading a new revised version of your manuscript as soon as possible.

Yours sincerely,

Poonam Bheda, PhD
Scientific Editor
Molecular Systems Biology

Please click on the link below to submit your revision:

-Editor's requests:

1) In the main manuscript file, please reduce keywords to max. 5.

Done, we retained the five most relevant ones

2) Please format the Data availability section describing how the data, code etc. have been made available according to the example below:

"The datasets and computer code produced in this study are available in the following databases:

- Chip-Seq data: Gene Expression Omnibus GSE46748

(<https://www.ncbi.nlm.nih.gov/geo/query/acc.cgi?acc=GSE46748>)

- Modeling computer scripts: GitHub (<https://github.com/SysBioChalmers/GECKO/releases/tag/v1.0>)

- [data type]: [full name of the resource] [accession number/identifier] ([doi or URL or identifiers.org/DATABASE:ACCESSION])"

The data availability section has been modified as requested. As for the last point, since we used a large number of publicly available datasets (189) we included them in Dataset EV4

3) Please ensure that the sequencing dataset GSE219240 is now publicly available.

The dataset has been made available

4) Please rename "Competing interests" to "Disclosure and competing interests statement". We updated our journal's competing interests policy in January 2022 and request authors to consider both actual and perceived competing interests. Please review the policy <https://www.embopress.org/competing-interests> and update your competing interests if necessary.

The title of this paragraph has been changed and the policy reviewed.

5) Author contributions: Please remove it from the manuscript and specify author contributions in our submission system. CRediT has replaced the traditional author contributions section because it offers a systematic machine-readable author contributions format that allows for more effective research assessment. You are encouraged to use the free text boxes beneath each contributing author's name to add specific details on the author's contribution. More information is available in our guide to authors: <https://www.embopress.org/page/journal/17574684/authorguide#authorshipguidelines>

The author contributions have been removed from the manuscript file and specified in the online system

6) Please check the "Author Checklist" carefully and complete all relevant questions. Currently information about animals (mice) is missing in the "Experimental animals" section.

We have now included information related to the animals used in the study

7) In the Materials and Methods, please take care of the following:

- Please ensure that a statement on whether or not blinding was done is included in the Materials and Methods even if no blinding was done. Please also ensure this part of the Author Checklist is updated.

- Animals: Please ensure that housing conditions as husbandry of the animals involved in experiments is reported. Please also ensure this part of the Author Checklist is updated.

- Antibodies: Please ensure that dilutions/amounts of each antibody used are reported in the relevant Materials and Methods sections.

-A statement of blinding has now been included at the beginning of the Materials and Methods section and the Author Checklist integrated accordingly

-Information related to mouse housing and husbandry has been included and reported in the Author checklist

-Information on antibody usage has been included

8) Please place individual sections of the manuscript in the following order: Title page - Abstract & Keywords - Introduction - Results - Discussion - Materials & Methods - Data Availability - Acknowledgements - Disclosure and Competing Interests Statement - References - Figure Legends - Expanded View Figure Legends.

The sections of the manuscript has been reordered as indicated

9) For the figures and figure legends, please take care of the following:

- Please define the annotated p-values *****/*** in the legend of figure 2h. Please note we require exact p-values to be reported.
- Please indicate the statistical test used for data analysis in the legends of figures 2h; 4f-g; 5a, c-f; 6d.
- Please note that the box plots need to be defined in terms of minima, maxima, centre, bounds of box and whiskers, and percentile in the legends of figures 2d, f, h; 6f.
- Please note that information related to n is missing in the legends of figures 2d, h; 5c; 6e.
- Please note that for heatmap present in figure 2b; a numbered scale bar is not provided. This needs to be rectified.
- Expanded View Dataset legends should be removed from the main manuscript file and rather each one should be individually included as a separate tab in the relevant Excel file.
- Appendix info ("Appendix/Appendix Figures S1 to S7") should be removed from main manuscript file.

-The indicated figure legends as well as the one of figure 4e have been amended and integrated as indicated.

-EV dataset legends have been removed from the manuscript text and integrated in the excel files.

-Appendix info has been removed from the manuscript file

10) Please ensure that all funding sources are entered into the manuscript submission system. "This work was also partially supported by the Italian Ministry of Health with the "Ricerca Corrente" and "5x1000" funds to the IEO IRCCS." is inserted in the comments box, but needs to be included in the "More Funders" list. A fellowship from the Associazione Italiana Ricerca sul Cancro (AIRC) is also missing from the manuscript submission system.

The funding sources have been amended as indicated

11) Synopsis:

- Synopsis image: Please upload as a high-resolution jpeg file and ensure that the dimensions are 550 pixels wide x (250-400) pixels high.

- Synopsis text: Please shorten the standfirst (maximum of 300 characters, including spaces) and limit it to 1 sentence.

Synopsis image and text have been modified to comply with the MSB guidelines

12) Please upload the Appendix as a single PDF (no separate image files are needed).

The Appendix has been uploaded as a single pdf file

13) Please ensure that a completed Source Data checklist (sent to you by Hannah Sonntag) is uploaded.

The source data checklist has been uploaded

14) As part of the EMBO Publications transparent editorial process initiative (see our policy here: https://www.embopress.org/transparent-process#Review_Process), Molecular Systems Biology will publish online a Peer Review File (PRF) to accompany accepted manuscripts. This file will be published in conjunction with your paper and will include the anonymous referee reports, your point-by-point response and all pertinent correspondence relating to the manuscript. Let us know whether you agree with the publication of the PRF and as here, if you want to remove or not any figures from it prior to publication. Please note that the Authors checklist will be published at the end of the PRF.

We agree with the publication of the PRF

18th Apr 2024

Manuscript number: MSB-2023-12088RRR

Title: An integrative epigenome-based strategy for unbiased profiling of clinical kinase inhibitors

Dear Dr. Natoli,

Thank you again for sending us your revised manuscript. We are now satisfied with the modifications made and I am pleased to inform you that your paper has been accepted for publication.

Yours sincerely,

Poonam Bheda, PhD
Scientific Editor
Molecular Systems Biology
